# Recent Approaches for the Topical Treatment of Psoriasis Using Nanoparticles

**DOI:** 10.3390/pharmaceutics16040449

**Published:** 2024-03-25

**Authors:** Krisztina Bodnár, Pálma Fehér, Zoltán Ujhelyi, Ildikó Bácskay, Liza Józsa

**Affiliations:** 1Department of Pharmaceutical Technology, Faculty of Pharmacy, University of Debrecen, Nagyerdei körút 98, 4032 Debrecen, Hungary; bodnar.krisztina@pharm.unideb.hu (K.B.); feher.palma@pharm.unideb.hu (P.F.); ujhelyi.zoltan@pharm.unideb.hu (Z.U.); bacskay.ildiko@pharm.unideb.hu (I.B.); 2Doctoral School of Pharmaceutical Sciences, University of Debrecen, Nagyerdei körút 98, 4032 Debrecen, Hungary; 3Institute of Healthcare Industry, University of Debrecen, Nagyerdei körút 98, 4032 Debrecen, Hungary

**Keywords:** psoriasis, nanoparticles, topical therapies, novel delivery systems, nanocarriers

## Abstract

Psoriasis (PSO) is a chronic autoimmune skin condition characterized by the rapid and excessive growth of skin cells, which leads to the formation of thick, red, and scaly patches on the surface of the skin. These patches can be itchy and painful, and they may cause discomfort for patients affected by this condition. Therapies for psoriasis aim to alleviate symptoms, reduce inflammation, and slow down the excessive skin cell growth. Conventional topical treatment options are non-specific, have low efficacy and are associated with adverse effects, which is why researchers are investigating different delivery mechanisms. A novel approach to drug delivery using nanoparticles (NPs) shows promise in reducing toxicity and improving therapeutic efficacy. The unique properties of NPs, such as their small size and large surface area, make them attractive for targeted drug delivery, enhanced drug stability, and controlled release. In the context of PSO, NPs can be designed to deliver active ingredients with anti-inflammatory effect, immunosuppressants, or other therapeutic compounds directly to affected skin areas. These novel formulations offer improved access to the epidermis and facilitate better absorption, thus enhancing the therapeutic efficacy of conventional anti-psoriatic drugs. NPs increase the surface-to-volume ratio, resulting in enhanced penetration through the skin, including intracellular, intercellular, and trans-appendage routes. The present review aims to discuss the latest approaches for the topical therapy of PSO using NPs. It is intended to summarize the results of the in vitro and in vivo examinations carried out in the last few years regarding the effectiveness and safety of nanoparticles.

## 1. Introduction

Psoriasis (PSO) is a relatively common chronic skin condition that affects people of all ages, ethnicities, and genders. There is a genetic component to psoriasis, and individuals with a family history of the condition are at a higher risk. According to the scientific literature, several genes related to the regulation of the immune system and the growth of skin cells play a significant role in the development of the disease. However, PSO is a multifactorial condition, as certain other factors can trigger or exacerbate psoriasis symptoms. These include stress, infections, and injuries to the skin. Moreover, several medications have been associated with PSO onset, as well as the exacerbation of the condition. The most reported active ingredients known to trigger PSO are lithium, beta-blockers, antimalarials, tetracyclines, and non-steroidal anti-inflammatory medications [1,2,3]. Identifying and managing these triggers can help in controlling the condition. It is estimated that approximately 2–3% of the global population is affected by PSO and an estimated 30% of patients with psoriasis also develop psoriatic arthritis [4,5]. The condition can develop at any age, but it most commonly appears for the first time between the ages of 15 and 25 [6].

It is important to note that psoriasis is a chronic condition with no cure, and its course can be unpredictable. Some individuals may experience periods of remission, where symptoms improve or disappear, while others may have persistent symptoms. Proper management and treatment, often tailored to the specific needs of each individual, can help control symptoms and improve the quality of life of those affected by psoriasis.

Psoriasis treatment typically aims to reduce inflammation, slow down the growth of skin cells, and alleviate symptoms. The choice of treatment depends on the severity of the condition, the type of psoriasis, and individual factors such as age and overall health [7]. Topically applied active ingredients are often the first line of defense for mild to moderate PSO. They include corticosteroids, vitamin D analogs, retinoids, coal tar, dithranol, and calcineurin inhibitors [8]. However, the topical treatment also comes with several challenges. The anti-psoriatics may not penetrate the skin deeply enough to effectively treat thicker plaques or those located on the palms, soles, or scalp. Because of their penetration-enhancing effect, nanoparticles have gained interest in the field of dermatology, including in the treatment of PSO [9]. While research is still ongoing, nanoparticles offer potential benefits for the management of psoriasis due to their unique properties, such as their small size, large surface area, and ability to encapsulate and deliver drugs to targeted areas. By delivering the anti-psoriatic drugs directly to the affected skin, nanoparticles may improve drug penetration, increase efficacy, and reduce side effects associated with systemic absorption. They can be incorporated into various formulations, including creams, gels, ointments, and sprays, to improve the stability, solubility, and bioavailability of drugs used to treat PSO. These formulations may enhance patient compliance and comfort [9,10]. Several types of liposomes, solid lipid nanoparticles (SLNs), nanoemulsions (NEs), nanostructured lipid carriers (NLCs), niosomes, nanosponges, ethosomes, dendrimers and nanocrystals containing anti-psoriatics have been written about in the scientific literature. While the potential of nanoparticles in PSO treatment is promising, more research is needed to optimize their use, ensure safety, and determine long-term efficacy. Additionally, regulatory approval and commercialization of nanoparticle-based therapies for PSO may require further validation through clinical trials.

The main purpose of the present review is to summarize the latest research based on the nanoparticles mentioned above, which have been formulated for the better management of PSO. It aims to discuss the unique advantages of these nanocarriers and the results of in vitro, ex vivo or in vivo studies. Furthermore, we present the latest clinical trials on nanoparticles for the treatment of PSO.

## 2. Characterization and Pathophysiology of Psoriasis

The skin is the largest organ in the human body and is composed of three main layers. The outermost layer, known as the epidermis, is primarily composed of specialized cells called keratinocytes, forming four identifiable layers: the stratum basale, stratum spinosum, stratum granulosum, and stratum corneum. The next layer is the dermis, which mainly comprises a fibrous extracellular matrix (ECM) produced by nearby fibroblasts along with immune cells dispersed throughout. The deepest layer, the hypodermis, primarily consists of adipose tissue [11]. These layers form a crucial network that serves as the body’s primary defense against pathogens, ultraviolet (UV) radiation, chemicals, and physical damage. Additionally, the skin plays a vital role in temperature and water balance regulation. The skin barrier consists of two main components: a physical barrier primarily located in the epidermis and an immune barrier found in both the dermis and epidermis. These two systems work together to preserve skin homeostasis and promote overall human health. However, if these mechanisms become disrupted, psoriasis may develop [12]. 

As mentioned above, the pathomechanism of psoriasis involves a complex interplay of genetic, immunologic, and environmental factors. PSO is considered an autoimmune disorder—without any obvious immunogen identified—as T cells are activated and migrate to the skin, where they trigger an inflammatory response [13,14]. Activated T cells release proinflammatory cytokines, including tumor necrosis factor-alpha (TNF-α), interleukin-17 (IL-17), and interleukin-23 (IL-23) [15]. These cytokines stimulate the proliferation of keratinocytes in the epidermis. Inflammatory cells, including neutrophils and dendritic cells, are recruited to the epidermis, leading to redness and swelling. The increased number of immune cells in the skin contributes to the characteristic plaques and lesions of PSO. Moreover, this condition is associated with increased blood vessel formation (angiogenesis) in the skin, which supports the increased metabolic demands of rapidly proliferating skin cells and contributes to the redness and warmth observed in psoriatic lesions [16]. The interaction between immune cells, cytokines, and keratinocytes creates a self-perpetuating cycle of chronic inflammation, which can lead to the persistent nature of PSO [17]. The pathomechanism of the disease is presented on Figure 1 (created with BioRender.com).

Understanding these mechanisms has led to the development of targeted therapies that specifically block key cytokines involved in the inflammatory cascade. These treatments aim to interrupt the pathomechanism at various points, providing more effective and targeted management of psoriasis symptoms. In addition, ongoing research continues to advance our understanding of psoriasis and potentially reveal new therapeutic targets and approaches.

The most common type of the disease is the plaque psoriasis (psoriasis vulgaris), which usually presents as well-defined, raised, red patches with silvery–white scales. Psoriasis vulgaris often occur on the elbows, knees, scalp, lower back, and nails, and it is characterized by excessive skin cell turnover, leading to the accumulation of dead skin cells on the surface. This results in scaling, which is a hallmark feature of the condition [18]. Another subtype is the inverse psoriasis, which is also called hidden psoriasis or intertriginous psoriasis. It affects skin folds, such as the neck, buttocks, under your arms, under your breasts, or in the groin or inner thigh area. The affected skin is typically red, shiny, and inflamed due to the increased blood flow and immune cell infiltration. Moreover, psoriatic lesions can be itchy and, in some cases, painful. PSO can affect the nails, causing pitting, ridges, and discoloration. In severe cases, it may lead to nail detachment. If the condition affects the scalp, forehead or back of the neck, we can talk about scalp PSO [19,20]. A further type of the disease is the so-called “guttate psoriasis”, which is characterized by small, teardrop-shaped papules that often appear after a streptococcal throat infection. This type of PSO is more common in children and young adults [21]. Rare forms of the disease are the pustular and the erythrodermic PSO; the first can be characterized by the quick development of pus-filled blisters, while in the case of the other type, the entire skin surface becomes red and inflamed, which itches and burns intensely [22]. It is worth considering that some patients may experience a combination of these types, and the severity can vary widely. The applied treatment depends on the type and the severity of the disease. Figure 2 presents the main types of the disease (created with BioRender.com).

## 3. Conventional Topical Treatment of Psoriasis

Topical treatment is generally considered a first-line treatment in mild to moderate psoriasis. However, creams, gels and ointments containing anti-psoriatic active agents are also used in the case of moderate to severe psoriasis, but they are usually combined with other treatments, such as phototherapy or systemic therapy [23]. Corticosteroids are a common and effective class of medications used to manage psoriasis symptoms as they suppress inflammation and immune responses in the skin. Topical corticosteroids, which are applied directly to the skin, are the primary form of corticosteroid treatment for psoriasis. Several potencies are available, ranging from Class 1 (highest potency) to Class 7 (Table 1). They are often recommended for short-term use (a few weeks and then a break or a less potent formulation is recommend) to avoid potential side effects associated with long-term use, such as skin thinning, skin atrophy, striae (stretch marks), and telangiectasia (dilated blood vessels). It has been described in several articles that the vehicle can directly modify the therapeutic and adverse effects of an active substance by changing its pharmacokinetics as well as the structure of the skin. Therefore, a priority aspect of dermatological and pharmaceutical research is the development of a suitable carrier for corticosteroids [24,25]. For example, Bhutani and her research group have reported that clobetasol incorporated into a spray vehicle proved to be more efficacious than other vehicles. However, this better efficacy could also occur because of the higher patient compliance with an odorless, easy-to-apply vehicle [25].

Several articles have described the anti-psoriatic activity of the topically used retinoids (derivatives of vitamin A); however, they are not typically considered first-line treatments for PSO. They can be used effectively for managing psoriasis on the face or in skin folds—for example, in the case of inverse PSO—where the skin is thinner and more sensitive. Topical retinoids, such as tazarotene and acitretin, act by regulating the turnover of keratinocytes, as they can influence the differentiation and proliferation, helping to normalize the growth and shedding of the epidermal cells. Weinstein et al. have found that the application of 0.1% and 0.05% tazarotene creams resulted in a significant reduction in the severity of PSO. They also found that by increasing the concentration of the active ingredient, the effectiveness increased, but more side effects appeared [26]. Overall, despite their valuable properties, topical retinoids can cause many side effects, such as skin irritation, photosensitivity, redness, and peeling, especially when treatment is initiated, which may limit their use in PSO. 

Topical calcineurin inhibitors also have an important role in the treatment of PSO, as they can modulate the immune system and have anti-inflammatory effects. While they are more commonly associated with the treatment of atopic dermatitis (eczema), they can also be used in certain cases for the management of psoriasis, particularly in sensitive areas (e.g., facial and genital PSO) where other treatments are less suitable because of the increased percutaneous penetration and, consequently, increased risk of adverse effects [7,27]. The two principle calcineurin inhibitors used in dermatology are tacrolimus and pimecrolimus, which act by the inhibition of the activity of calcineurin, an enzyme involved in the activation of T cells. By suppressing T cell activation, these medications help to reduce inflammation and the abnormal immune response that characterizes PSO [27].

Nevertheless, several clinical trials comparing topical corticosteroids with topical calcineurin inhibitors have found that topical corticosteroids are equally or more effective than topical calcineurin inhibitors and have fewer adverse effects [27,28]. Kreuter and his research group described that the 0.1% betamethasone formulation proved to be more effective than the formulation containing 1% pimecrolimus in the treatment of inverse PSO. It has also been suggested that for patients with inverse psoriasis, intermittent therapy with short-term topical corticosteroids followed by maintenance treatment with a less potent agent, such as pimecrolimus or calcipotriol (CPT), may be appropriate. [29].

Vitamin D analogs, such as calcipotriene, are commonly used in the treatment of PSO, as well. In addition, novel vitamin D analogs, including maxacalcitol, paricalcitol and becocalcidiol, are being studied for the treatment of PSO [30,31]. They exert their effects on the affected skin by modulating the differentiation of keratinocytes and inhibiting epidermal hyperproliferation, helping to normalize the excessive growth seen in PSO plaques [32]. They are often used in combination with other topical treatments, such as corticosteroids, to enhance their effectiveness, as they act via distinct and complementary mechanisms of action. In addition, certain studies have also demonstrated that vitamin D may be able to restore epidermal barrier function damaged by corticosteroid use and reduce the chance of developing steroid-induced skin atrophy [32,33,34]. This is why corticosteroids and vitamin D analog combination topical treatment can provide the more comprehensive and long-term management of PSO [35].

The topical use of anthralin (1,8-dihydroxy-9anthrone, dithranol) has anti-proliferative effects on skin cells, as well [36]. This active ingredient was first synthesized as a derivative of chrysarobin, prepared from the araroba tree in Brazil over a century ago [37]. It has been part of psoriasis therapy for years, as it helps slow down the excessive growth and multiplication of keratinocytes. This is why it is particularly effective in reducing the thickness of psoriasis plaques, although the staining and irritating effects significantly limit the use of this active agent. Several studies have suggested that dithranol may be effective in high concentrations for short periods of skin contact, while a few studies support the use of dithranol combined with other topical treatments or ultraviolet radiation B (UVB) phototherapy to improve the response in PSO of the body [38,39,40,41].

Salicylic acid plays a role in the treatment of psoriasis through its combined anti-inflammatory and exfoliating effects. It effectively reduces scaling, induration, itching, and erythema and promotes the shedding of scales [8]. It can also enhance the penetration and, thus, the bioavailability of other anti-psoriatics, such as corticosteroids or vitamin D analogs, by thinning the outer layer of the skin [42].

Applying nonmedical moisturizers and emollients before topical medications can also help soften psoriatic plaques and improve the penetration of the above-mentioned drugs. They hydrate the skin, minimize cracking and flares, and alleviate symptoms such as itching and dryness. While they may not directly treat the inflammation associated with PSO, they contribute to overall skin health and can complement other treatments.

Overall, the conventional topical therapy for PSO can be effective in managing the condition, but it also has its disadvantages. While topical treatments can be beneficial for mild to moderate PSO, they may not be potent enough to effectively treat severe cases. The main problem with topical medications is that they can cause skin irritation, including redness, itching, or burning. This can be particularly problematic for individuals with sensitive skin or if the treatment is applied to areas with open wounds. For example, calcineurin inhibitors and vitamin D analogs, especially calcipotriene, tazarotene and dithranol, can cause skin irritation on or around the psoriasis plaques. The other huge problem is the low permeability of these active ingredients [30]. This is why there is a great demand for the use of innovative, nano-sized drug carrier systems.

## 4. Challenges in the Topical Delivery of Anti-Psoriatics

The topical delivery of anti-psoriatics faces several challenges that impact the effectiveness and patient compliance of treatment. These challenges include issues related to the characteristics of the skin, drug formulation, and patient-specific factors [43].

The outermost layer of the skin, called the stratum corneum, serves as a barrier that limits the penetration rate of the active ingredients. Psoriasis often involves thickened and hyperkeratotic plaques, further hindering drug penetration, which results in the ineffective delivery and efficacy of the conventional treatment. In the conventional medication formulation approach, for topical drugs, it is difficult to break through the outermost layer of the skin [43,44,45]. This problem is further complicated by extreme disease-associated conditions such as hyperkeratosis and irritation [46]. The increased cholesterol level and reduced ceramide content of the skin characteristics of psoriasis can further complicate the penetration of the active ingredients into the deeper layers of the skin. In addition, hydrating stimuli such as skin water are inadequate, limiting the entry of relatively large amounts of the drug. However, the level of hydration of the skin is not adequate in most cases, which limits the penetration of the anti-psoriatics to a relatively large amount [47]. Furthermore, skin permeability varies across different body regions. For example, the skin on the face and genitals is more permeable than that on the palms and soles. The choice of a delivery system must consider the specific characteristics of the affected area. Some anti-psoriatics drugs belong to the Biopharmaceutical Classification System (BCS) Class II (e.g., tacrolimus, dithranol and calcipotriene), denoting that they have poor water solubility, which may affect their absorption and bioavailability. This is why formulations that enhance drug solubility are often required [48].

Topical treatments for psoriasis typically require consistent and long-term application. Patient compliance can be challenging due to factors such as the inconvenience of frequent application, the time-consuming nature of treatment, and potential side effects. As mentioned above, prolonged use of anti-psoriatics, especially corticosteroids with potent effects, may cause skin irritation, dryness, or itching, which can lead to reduced patient adherence and discontinuation of treatment. Furthermore, topically applied drugs can be absorbed into the bloodstream, as well, if applied to large areas of the body or in patients with impaired skin integrity. This may increase the risk of systemic side effects [49,50].

## 5. Nanotechnology and Nanomedicines

Nanotechnology is a multidisciplinary field of science that involves the synthesis, characterization and manipulation of a given material at the nanoscale, typically less than 100 nanometers (nm). At this scale, the properties of the substances can differ significantly from those at the macroscopic level, leading to novel properties. Polymer scientists often refer to polymer drug delivery systems as “drug nanocarriers”, even though their size can exceed 100 nm. This term reflects their primary function of carrying drugs within a nanoscale structure, facilitating targeted delivery, and improving drug efficacy. The application of nanotechnology extends to various industries; typically, science, engineering and technology disciplines take advantage of its benefits [51,52].

Nanomedicine refers to the application of nanotechnology in the field of medicine for the diagnosis, treatment, and prevention of diseases. The use of nanoscale substances and devices in medicine has the potential to revolutionize healthcare by providing more targeted and effective approaches to various medical challenges. Nanomedicines can take different forms, including nanoparticles, nanocarriers, and nanodevices, and they offer unique properties that can be exploited for therapeutic and diagnostic purposes, as well [53]. 

In the past few decades, the science of nanotechnology has attracted great interest from researchers in various fields, including drug delivery. The use of nanotechnology in medicine offers many advantages over conventional drug delivery systems, such as modifying the solubility of hydrophobic drugs or promoting stability. In the development of a nanocarrier-mediated drug delivery system, critical factors such as the surface charge (zeta potential), size, and morphology of nanoparticles, as well as the entrapment efficiency and drug permeation rate, are essential parameters to be meticulously addressed for the formulation of a successful and efficient product [54,55]. Nanoparticles can be designed to deliver drugs specifically to the site of disease, thus minimizing systemic side effects, achieving controlled or sustained release, and improving therapeutic efficacy. Several studies concluded that nanoparticles can deliver even genetic material, such as DNA or RNA, for gene therapy applications. The recently approved pioneering mRNA vaccines (e.g., BioNTech/Pfizer (Comirnaty) and Moderna mRNA-1273), which use lipid-based nanoparticles (LNPs) as carriers of mRNA encoding the SARS-CoV-2 spike glycoprotein, have also demonstrated the importance of nanoparticles in gene therapy [56,57]. Nanocarriers can also modify the pharmacokinetics of the active agents, extending their circulation time and improving their bioavailability.

The skin is a barrier to drug penetration due to the structure of the outermost layer of the epidermis, the stratum corneum. Permeation through an intact epidermis is very difficult for molecules larger than 200–350 Daltons [58]. Several studies have explored the potential of nanosized carriers for topical drug delivery, recognizing that their behavior differs when interacting with biological barriers [59,60,61]. This is why nanotechnology could provide an innovative and promising new treatment option for several dermatological diseases, such as PSO. Given that the lipids of the stratum corneum play a significant role in creating a barrier to penetration, lipid carriers, which facilitate lipid exchange with the skin surface, show considerable promise [62]. They can interact with the lipid-rich stratum corneum, facilitating the penetration of drugs across the barrier. The most commonly applied lipid nanoparticles include liposomes, solid lipid nanoparticles (SLNs), nanostructured lipid carriers (NLCs) and nanoemulsions [60,63,64,65] (shown in Figure 3). 

Lipid nanocarriers can improve the solubility of poorly water-soluble drugs, allowing for better incorporation and delivery of a wider range of therapeutic agents. Moreover, they can protect drugs from degradation, oxidation, or hydrolysis, thereby improving the stability of the encapsulated active ingredient. In addition to increasing the bioavailability, they provide controlled and sustained release of the active agent, leading to prolonged therapeutic effects and potentially reducing the frequency of application. Lipid nanocarriers allow for the incorporation of multiple drugs, enabling combination therapy for synergistic effects in treating complex skin conditions, such as PSO [64,66,67,68]. These nanoparticles consist of physiological lipids and are therefore well-tolerated because of their degradation into nontoxic residues. These nanocarriers are formulated using a range of biodegradable lipids that share structural similarities with the components found in the epidermis, particularly in the stratum corneum (SC) [69,70]. 

Besides lipid nanocarriers, polymer particles can enhance dermal uptake or improve tolerability, as well. Nanosized polymeric carriers have well-controllable sizes and surface polarity, allowing adaptation to the intended target. Polymeric nanoparticles are composed of biodegradable polymers, such as poly(lactic-co-glycolic acid) (PLGA) or chitosan. In the case of polymeric micelles, they are composed of amphiphilic block copolymers that self-assemble into nanoscale micelles, which can enhance the solubility of hydrophobic drugs. Another group of polymeric nanoparticles is the dendrimers, which are hyperbranched polymers with well-defined structures that allow the controlled drug loading and release of both hydrophilic and hydrophobic drugs [71]. There is certain research indicating the development of a nano-hydrogel through the encapsulation of drugs, therapeutic proteins, or vaccine antigens using hydrophobic polysaccharides [47,72,73].

However, it is worth noting that while nanotechnology offers enormous opportunities in various fields, it also comes with many limitations. Certain nanoparticles may exhibit toxic effects, depending on their composition, size, surface properties, and route of administration. It is reported in several studies that nanoparticles can be distributed to the liver, heart, spleen, and brain [74], which is why scientists are concerned about the unpredictable health outcomes of nanoparticles due to their distinctive physicochemical properties in various biological systems. Furthermore, they can also cause allergic reactions because of their strong immunomodulating properties [75]. Another major concern is the accumulation and the lack of biodegradation. Nanoparticles intended for biomedical applications, such as drug delivery or imaging agents, must be designed to degrade or be cleared from the body after fulfilling their intended function. Ensuring appropriate biodegradability and elimination pathways is vital to prevent long-term accumulation and potential adverse effects [76]. The development and application of products based on nanotechnology can incur significant expenses due to the specialized equipment, materials, and expertise necessary. Elevated production costs might restrict the extensive utilization of nanotechnology in particular industries or contexts, thereby hindering accessibility and affordability. Biological challenges, large-scale manufacturing, biocompatibility and safety, government regulations, and overall cost-effectiveness compared to current therapies must be considered in the clinical development of nanoparticles. These factors may present significant barriers to its occurrence, regardless of whether they are therapeutically beneficial or not [77]. To tackle these challenges, interdisciplinary cooperation and risk evaluation are necessary. 

## 6. Current Nanotechnology-Based Approaches for the Topical Treatment of Psoriasis

Nanospheres such as liposomes, nanostructured lipid carriers (NLCs), niosomes, solid lipid nanoparticles (SLNPs), nanocrystals, ethosomes, nanosuspensions and dendrimers are being actively investigated for their potential in the targeted treatment of PSO. The choice of nanoparticle depends on factors such as the specific characteristics of the drug, the desired mode of action, and the properties of the skin. 

### 6.1. Liposomes

Research in the field of psoriasis treatment is increasingly intensive and various approaches have been investigated, including liposomes. Liposomes are lipid-based, nano-sized bilayer structures that exist in various forms, including small unilamellar vesicles (SUVs), large unilamellar vesicles (LUVs), multilamellar vesicles (MLVs), and multivesicular vesicles (MVVs), depending on their structure and composition [78]. They have gained wide acceptance as nanocarriers for both hydrophilic and lipophilic drugs due to their high biocompatibility, biodegradability, and minimal toxicity or immunogenicity [79]. Liposomes can enhance drug solubility, regulate drug distribution, and offer the flexibility of surface modification to achieve target-specific sustained release [80]. Typically, they are used for drug encapsulation, ranging in size from 50 to 150 nm, making them suitable for drug delivery by various routes. In the context of psoriasis, liposomal formulations have been investigated to enhance the targeted delivery of therapeutic agents to the skin, potentially improving efficacy and reducing side effects.

The primary issue encountered when using liposomes lies in their limited physical and chemical stability, attributed to the delicate nature of phospholipid membranes and their susceptibility to peroxidation. Physical degradation may also occur because of alterations in the structure of the liposomes [81].

Mezei and Gulasekharam were pioneers in reporting the dermal delivery of drugs in liposomal form. They conducted a comparative study on the retention of a corticosteroid, the triamcinolone acetonide in the epidermis and dermis, using a liposomal lotion and gel, in contrast to conventional lotion and gel formulations of the free drug. The findings demonstrated that the liposomal gel exhibited a five-fold increase in retention in the epidermis and a three-fold increase in the dermis compared to the free drug in conventional formulations [82]. Another study on triamcinolone acetonide focused on the development and characterization of multilamellar liposomes. These liposomes were created using varying compositions of L-alpha-phosphatidylcholine and the drug, with investigations into different storage conditions. The study aimed to assess the encapsulation efficiency and drug loss. Stability assessments revealed that refrigeration (4–6 °C) enhanced stability, reducing early diffusion of the drug through the liposomal wall compared to room temperature storage. The incorporation of cholesterol in some formulations improved stability under both refrigerated and room temperature conditions; however, a decrease in the encapsulation efficiency occurred compared to formulations without cholesterol. Consequently, antioxidants and/or preservatives were introduced to achieve desirable vesicle dimensions, optimal encapsulation efficiency, and improved stability [54,83]. 

The use of all-trans-retinoic acid and betamethasone for PSO treatment is still challenging due to their poor stability, limited skin permeation, and potential systemic side effects. Therefore, Wang et al. endeavored to devise a dual-loaded flexible liposomal gel to enhance the therapeutic efficacy of psoriasis treatment with all-trans-retinoic acid and betamethasone. Compared to free drugs, these liposomes significantly enhanced the skin permeation and retention of the above-mentioned drugs [84].

Bexarotene functions, as a retinoid X receptor agonist, were initially approved for anticancer purposes in 1999 and later identified as having anti-psoriasis properties in several studies. Nevertheless, the poor aqueous solubility and high log P present challenges in achieving topical delivery. Therefore, Saka et al. attempted to resolve this issue by formulating bexarotene into a cholesterol-containing liposome. An imiquimod (IMQ)-induced psoriasis plaque model in BALB/c mice verified the successful reversal of psoriasis through liposomal bexarotene. This was demonstrated by a decrease in scaling and inflammation without any toxicity. In addition, subsequent histopathological and cytokine level changes served to confirm the recovery of the animals treated with the liposomal gel preparation. This suggested that liposomal bexarotene could be a compelling clinical contender for the management of psoriasis, assuming it is subject to clinical evaluation in a suitably designed and validated trial [85]. In another study, Knudsen et al. explored the impact of incorporating the lipopolymer polyethylene glycol-distearoyl phosphoethanolamine (PEG-DSPE) for stabilizing liposomes, investigating its effects on the liposomal physicochemical properties and the delivery efficiency of membrane-intercalated calcipotriol into the skin. The addition of 0.5, 1, and 5% PEG-DSPE to the membrane improved the colloidal stability of the liposomes while maintaining the effective delivery of calcipotriol into excised pig skin. Notably, liposomes loaded with calcipotriol and containing 1% PEG-DSPE demonstrated a significant increase in calcipotriol deposition into the stratum corneum. The size of the liposomes played a role in calcipotriol penetration, with SUVs facilitating greater penetration compared to LUVs. The study found that calcipotriol penetrated the skin more effectively than the lipid component of the liposomes, indicating that a fraction of the drug was likely released from the liposomes during skin migration. In summary, PEGylation emerged as a promising strategy for enhancing the stability of calcipotriol-loaded liposomal formulations without compromising their skin accumulation properties [86]. Walunj et al. developed and evaluated the efficacy of a topical liposomal gel containing cyclosporine-loaded cationic liposomal nanocarriers. The liposomes contained cholesterol and 1,2-dioleoyl-3-trimethylammoniumpropane (DOTAP), a cationic liposome-forming compound. The liposomal gels loaded with the drug exhibited shear-thinning behavior, making them suitable for topical application. They demonstrated a reduction in psoriasis symptoms and the levels of key psoriatic cytokines, including TNF-α, IL-17, and IL-22 [87]. 

Huang et al. prepared a cholesterol-free liposome loaded with ginsenoside Rg3 (a small molecular compound with anti-inflammatory and immune regulation functions) and successfully loaded the liposome into microneedles (MNs) with hyaluronic acid as the matrix. In vitro transdermal experiments showed that Rg3 delivered by MNs was retained in the skin for a long time and the bioavailability was improved. In addition, they also found that Rg3-MNs significantly reduced epidermal thickness and dermal papillary edema. A significant reduction was also found in the IL-17, IL-23 and TNF-α levels. The liposome system prepared and investigated in this study could also be loaded with other psoriasis drugs, such as methotrexate and calcipotriol [88]. A further study conducted by Javia and colleagues examined liposomes loaded with omiganan. Omiganan is a new synthetic cationic peptide with 12 amino acids from the cathelicidin family. The omiganan liposomal gel showed controlled release and a better permeation profile than conventional formulations. The liposomal product significantly reduced the levels of pro-inflammatory cytokines and improved the psoriatic lesions compared to the conventional gel and lotion-based formulations [89].

Celastrol is a naturally occurring triterpene compound derived from the roots of *Tripterygium wilfordii*, a traditional Chinese medicinal plant, whose anti-inflammatory properties have drawn interest in the context of inflammatory disorders, such as PSO [90]. Long Xi and colleagues developed celastrol-loaded liposomes that are mannosylated using the thin-film dispersion method to enhance drug uptake by the targeted cells. The dendritic cells exhibited a 2-3-fold increase in the uptake of drug-loaded mannose-grafted liposomes compared to the non-grafted formulation. Additionally, an improvement in the anti-maturation effect of the drug was discovered when it was encapsulated in the mannose-lipid-grafted liposome formulation in comparison to its free form. Therefore, this formulation can improve the uptake of celastrol within cells and its anti-maturation influence [91]. Another natural active ingredient, glabridin, is a principal component of isoflavonoids found in *Glycyrrhiza glabra*, exhibiting antibacterial, anti-inflammatory, and anticancer properties [92]. Lu et al. investigated the therapeutic benefits and mechanisms of glabridin liposome application in mice with IMQ-induced PSO. The experiments demonstrated a reduction of the Psoriasis Area and Severity Index (PASI) score of the mice, with a dose-dependent relationship. In addition, the inspection of pathological skin tissue sections revealed that these formulations alleviated multiple psoriatic symptoms triggered by IMQ, including epidermal dysplasia, mast cell infiltration, and degranulation. The enzyme-linked immunosorbent assay (ELISA) has also revealed that the liposomes decreased the production of TNF-α, IL-17, and IL-22 [93]. Jain and colleagues investigated the synergistic effect of a topical liposomal gel containing Ibrutinib and Curcumin for the management of psoriasis. The formulation was loaded into Carbopol gel and analyzed for its anti-psoriatic activity. The results of the in vitro release studies indicated that liposomes loaded with the Ibrutinib–Curcumin combination steadily released the contents for an extended period, offering benefits in circumventing frequent administrations. Additionally, it showed an increased level of safety, as demonstrated through skin compatibility testing [94].

However, it is important to note that while liposomal formulations show promise, their clinical effectiveness in PSO treatment may vary, and not all liposome-based treatments will necessarily be successful. Clinical trials are essential to evaluate the safety and efficacy of new treatments.

The recently formulated liposomes containing anti-psoriatic agents are listed and characterized in Table 2.

### 6.2. Solid Lipid Nanocarriers (SLNs)

Solid lipid nanoparticles (SLNs) are spherical particles composed of a solid lipid matrix in which drug molecules can be incorporated. Biocompatible lipids are utilized in producing SLNs that are safe for physiology, while a surfactant layer is used to stabilize the particles in the aqueous phase. SLNs possess the ability to transport drugs, vitamins, molecules, and almost all xenobiotics. It was demonstrated that they can improve solubility, cellular uptake, and stability while reducing enzyme degradation and prolonging the circulation period of various drugs [95]. 

Methotrexate (MTX) is a prevalent treatment for PSO due to its effectiveness. However, it possesses poor absorption characteristics, and there are various complications associated with the usage of this drug. This is why Maiti et al. employed various experimental models to assess the efficacy of solid lipid nanoparticles of MTX (SLNs-MTX) in treating PSO locally. The effectiveness of the SLNs-MTX preparation was compared to standard MTX and commercially available MTX preparations. The results indicated that MTX was gradually released from the formulation and permeated the skin entirely (80.36%). The formulation showed a dose-dependent suppression of keratinocyte expansion, and the lowest cytotoxic concentration recorded was 518 mcg/mL. As a result, it was deduced that MTX embedded in solid lipid nanoparticles could be a hopeful combination [96]. In a further recent study, SLNs-MTX formulated by a solvent diffusion technique has been investigated. It was found that the use of the nanoformulation resulted in a significantly higher deposition of MTX (71.52%) compared to the topical formulation containing only MTX (38.48%). In vivo studies revealed an enhancement of the therapeutic response and a decrease in local side effects when the SLNs-MTX-loaded formulation was used for the topical treatment of PSO. The anti-psoriatic effectiveness of the nanoparticles at a concentration of 100 µg/cm^2^ was also assessed, employing an IMQ-induced psoriasis model in BALB/c mice [97]. These studies reflected that using an NP formulation holds promise as an effective strategy to enhance the bioavailability, solubility, and circulation duration of a molecule with unfavorable topical bioavailability, while also facilitating its targeted delivery to specific tissues.

Trombino et al. aimed to enhance the absorption of topical cyclosporine while minimizing the side effects. To enhance the dermal penetration, they used SLNs as carriers due to their lipophilic and occlusive properties. The release profile of cyclosporine was evaluated, revealing site-specific release in skin layers and low transdermal release through in vitro testing. Moreover, SLNs demonstrated significant anti-inflammatory activity in vitro, proving to be effective in ameliorating the inflammatory condition that plays a role in the development of PSO [98]. Cyclosporine containing SLNs and NLCs was successfully developed by another research group using a hot homogenization method. According to their ex vivo skin delivery studies, the lipid nanoparticles increased the topical bioavailability of the peptide. It was also found that SLNs demonstrated reduced cytotoxicity when investigated on human keratinocytes compared to NLCs. Consequently, the application of SLNs proved to be the most favorable targeted system for locally delivering peptides to the skin [99]. 

Leflunomide (LEF), a drug that is used internally in adults with active rheumatoid arthritis and active psoriatic arthritis (arthritis associated with psoriasis), has been extensively examined for its potential to reduce inflammation in skin disorders like PSO. Nevertheless, topical administration presents difficulties due to the instability and skin irritation. Alhelal and his research group aimed to overcome these challenges by creating a hydrogel with LEF-loaded SLNs. They also prioritized enhancing the photostability and safety of LEF intended for topical application through the creation of LEF-filled solid lipid nanoparticles that were subsequently incorporated into a hydrogel. The results of this investigation suggested that the LEF-SLN hydrogel could improve the photostability of the trapped drug and alleviate skin irritation while possessing local delivery attributes [100].

Rapalli and colleagues aimed to minimize the systemic side effects by developing a hydrogel that is loaded with apremilast containing SLN. In vitro release studies disclosed that the drug was released for an extended period from the SLN dispersion. The MTT assay indicated that the excipients had minimal impact on the viability of cells, and the SLN dispersion was highly internalized. An ex vivo skin permeation and retention study with Coumarin-6 dye exhibited increased retention and permeation with the SLN formulation as compared to the gel containing the free drug. The outcomes intimate that apremilast-loaded SLN intended for topical use could be an effective therapy by specifically targeting the dermal layers. With clinical testing, the suggested formulation may potentially serve as an alternative for treating psoriasis in the near future [101].

Noscapine (NOS) exhibits numerous anti-inflammatory, anti-angiogenic, and anti-fibrotic properties, but its effectiveness in clinical settings is impeded by its limited solubility and size. To improve this, Rahmanian-Devin et al. developed SLNs encapsulating NOS (SLNs-NOS). When exposed to phosphate buffer pH 5.8 and 7.4, the release of NOS was 83.23% and 58.49%, respectively, after 72 h. The results of the Franz diffusion cell experiment showed that the concentration of NOS in the skin was 46.88% and 13.5% of the total amount for SLN and cream formulations, respectively. They also reported that the treatment with SLNs-NOS produced a significant reduction in the ear thickness and PASI score compared to the untreated group. Moreover, applying SLNs-NOS topically significantly reduced parakeratosis, hyperkeratosis, acanthosis, and inflammation when compared to the control group, according to histopathological studies [102]. The above-mentioned SLNs are summarized in Table 3.

### 6.3. Nanostructured Lipid Carriers (NLCs)

Nanostructured lipid carriers (NLCs) are innovative pharmaceutical formulations comprising physiologically and biocompatible lipids, surfactants, and cosurfactants. NLCs exhibit tremendous potential in the pharmaceutical and cosmetic markets due to their numerous advantageous effects, which include skin hydration, skin barrier, enhanced bioavailability, and skin targeting. To formulate NLCs, degradable and compatible lipids (solid and liquid) and emulsifiers are applied. Incorporating liquid lipids creates structural flaws in solid lipids, resulting in a less organized crystalline structure. This, in turn, prevents drug leakage and enables high drug loading. NLCs possess advantageous characteristics as drug delivery systems, including simple preparation, biocompatibility, the possibility of expansion in size, non-toxicity, and improved drug loading and stability [103].

In a study, Xu et al. developed a drug delivery system for PSO treatment in the form of an NLC gel containing luteolin. The NLC gel exhibited more effective permeation properties and a slower release rate of 36 h as compared to the gel containing the active ingredient without NPs. This study demonstrated that high dosages of the created gel (80 mg/kg body weight/day) decreased the levels of inflammatory and proliferative factors, including TNF-α, IL-6, IL-17, and IL-23, in both skin and blood [104].

Alam and his team developed a nanogel incorporating nanostructured lipid carriers loaded with tacrolimus and thymoquinone. Their research indicated that this nanogel formulation showed promise as a combined treatment for psoriasis. Moreover, it demonstrated the ability to reduce the toxicity typically associated with escalating dosages [105].

NLCs containing riluzole were synthesized, optimized, and characterized by Llorene et al. to address the challenges of topical therapies, particularly slow release, and improve stability. The in vitro release study demonstrated that NLCs facilitate extended and effective delivery of riluzole. Additionally, they exhibited non-angiogenic properties and impeded keratinocyte cell proliferation. Thus, the NLC containing riluzole and natural essential oils proved to be a favorable approach against hyperproliferative diseases of keratinocytes, such as PSO [106].

The study conducted by Morakul and colleagues aimed to encapsulate cannabidiol (CBD) extract in NLCs to enhance the chemical stability and anti-inflammatory effects of CBD for transdermal delivery. The CBD extract was loaded at 1%. They found that the encapsulation of CBD extract in NLCs successfully decreased its cytotoxicity in Human Dermal Fibroblasts and HaCaT cells. Further research is required to ensure appropriate clinical implementation and improve patients’ quality of life [107].

Thakur et al. formulated NLCs to incorporate tazarotene and calcipotriol. Their study focused on optimizing and characterizing the nano-lipid carrier formulation and integrating it into the Carbopol 934 gel base. An ex vivo investigation indicated minimal drug penetration into the systemic circulation, while a histopathological examination affirmed the potential advancement in the anti-psoriatic activity of the prepared NLCs. The tazarotene- and calcipotriol-filled NLC gel formulation demonstrated improved anti-psoriasis efficacy and sustained release when compared to the NLC gel containing only tazarotene [108].

As mentioned above, fluocinolone acetonide and acitretin are psoriasis medications that are widely used and classified under Class II and IV of the Biopharmaceutics Classification System [109,110]. Fluocinolone acetonide treatment can result in side effects like skin irritation and burning [109], whereas acitretin may have teratogenic effects and can cause xerophthalmia [111]. Raza et al. endeavored to generate topical NLCs to diminish the side effects and improve the therapeutic effectiveness. The ex vivo penetration study demonstrated lower permeation for the NLC gel co-formulated with fluocinolone acetonide and acitretin in comparison to the conventional fluocinolone acetonide and acitretin-containing gel. On the other hand, the skin deposition for the NLC formulation was greater than that of the conventional gel, verifying that NLCs were present in the deeper skin layers [112]. Table 4 summarizes the above-mentioned NLCs and their characteristics.

### 6.4. Nanoemulsions (NEs)

Nanoemulsions consist of two stable and transparent dispersions of insoluble oil and water phases, accompanied by surfactant particles ranging in size from 5 to 200 nm [113]. The utilization of nanoemulsions as a carrier for psoriasis drugs presents notable advantages compared to macroemulsions. These advantages include the elimination of common issues observed in macroemulsions, such as flaking, internal creaming, deposition, or coarsening [114]. When applied topically, nanoemulsions exhibit remarkable permeability and an enhanced capacity for drug loading. The achievement of an optimal nanoemulsion necessitates the careful selection of appropriate surfactants and oils [115]. 

In a recent study, Rai and his research group developed a nanoemulsion of tacrolimus and azelaic acid by applying soy lecithin and vitamin E oil to enhance topical drug availability and efficacy against plaque PSO. Carbopol 940 was used to concentrate the nanoemulsion to improve its residence time at the patient’s site. The gel exhibited a regulated drug release profile, and it exhibited a slow penetration rate of 24 h, which is anticipated to allow for consistent drug accessibility and absorption by the skin, ultimately leading to high skin retention of the drug for a longer duration compared to other tacrolimus ointment available on the market. An ex vivo study was conducted on freshly excised skin from pig ears to assess the skin permeation. Despite the water-based nature of the gel, it demonstrated considerable efficacy in maintaining skin moisture levels and delivering the drug accurately [116].

Several ongoing clinical trials indicate the potential effectiveness of tofacitinib in managing inflammatory skin conditions, such as PSO. Atmakuri and colleagues formulated tofacitinib citrate-loaded nanoemulsions as a topical solution by a spontaneous nanoemulsion technique. The concentration evaluated was 0.5% (*w*/*w*) of tofacitinib citrate loaded in a nanoemulsion gel formulation. An inflammation model was created using haptens like 2,4-dinitrochlorobenzene in BALB/c mice. ELISA was performed on skin homogenates to determine the levels of IL-6 and TNF-α. The results showed a noteworthy decrease in pro-inflammatory cytokines among animals treated with 0.5% tofacitinib citrate nanoemulsifier compared to the negative control group [117].

In their study, Jain et al. examined a nanoemulsion gel used to transport fluticasone propionate through the skin. The combined treatment, consisting of fluticasone propionate integrated with negative-entropy gel, babchi oil, and aloe vera gel, produced a synergistic effect in achieving long-lasting relief from PSO. According to the ex vivo analysis, the nanoemulsion gel ameliorated the skin permeability, leading to a nearly four-fold increase in fluticasone propionate retention in the deeper layers of skin. In vivo trials provided evidence that the nanoemulsion gel has superior anti-psoriasis effectiveness compared to the Flutivate marketed formulation [118]. In another study, a nanoemulsion hydrogel containing babchi oil as the sole active ingredient was produced. According to its results, the cumulative amount of babchi oil delivered through penetration and fluidization using the nanoemulsion gel was significantly higher compared to traditional formulations. The skin retention of the oil was found to be satisfactory, and the time delay decreased. The study concluded that the nanoemulsion demonstrated a significant improvement in drug penetration, suggesting its potential as a carrier for babchi oil delivery [119].

Khan et al. developed nanotherapeutic agents that incorporated bioactive compounds, such as thymoquinone and fulvic acid, into a nanoemulsion gel with calonji oil as the oil phase. To address the challenges of combining these natural bioactive agents, the researchers used the surfactant Tween 80 and co-surfactant Transcutol P in the formulation [120]. The permeability of the bioactive compounds was found to be 98.99%. Moreover, it exhibited a sustained dissolution rate of 75.76% and a permeation rate of 3.64 μg/cm^2^/h. In vivo studies with BALB/c mice as a model revealed the potent anti-inflammatory effect of thymoquinone, while the molecular docking studies confirmed the strong binding affinity of thymoquinone to both TNF-α and IL-6 receptors [121].

Algahtani et al. conducted a study aimed at improving the solubility and skin penetration of curcumin for topical application. To achieve this objective, the researchers optimized a curcumin nanoemulsion using a low-energy emulsification method and transformed it into a nanoemulsion. The permeation of curcumin exhibited a 4.87-fold increase compared to the simple curcumin gel. In an in vivo study using an IMQ-induced psoriasis model in male BALB/c mice, the efficacy of the curcumin-containing nanoemulsion gel was significantly superior to that of the curcumin gel without nanomaterials [122]. Khatoon et al. also conducted a study on the potential of natural bioactive substances, namely curcumin, resveratrol, and thymoquinone. However, the low aqueous solubility of these substances posed a challenge. In vivo experiments conducted in the BALB/c mouse model showed promising results in improving control of the disease. These findings support the potential effectiveness of the triple natural bioactive combination in the nanoemulsion gel formulation for the treatment of psoriasis [123].

The primary aim of the study conducted by Rashid and colleagues was to develop a nanoemulsion gel that contains methotrexate and olive oil. The release kinetics indicated that after 20 h, approximately 72.47% of the methotrexate was released at pH 5.5. The results of the Fourier Transform Infrared Spectrometer (FTIR) measurements suggested that the formulation has the potential to enhance the epidermis and dermis of the skin, thereby improving drug permeability and retention. Notably, the use of Tween 80 and PEG 400 as penetration enhancers significantly enhanced these effects. Permeation measurements revealed that, after 24 h, an average of 70.78 ± 5.8 μg/cm^2^ of methotrexate permeated from the nanoemulsion gel, with a flux value of 2.078 ± 0.42 μg/cm^2^/h [124].

Mittal and colleagues developed and optimized a nanoemulsion gel containing tacrolimus to improve the management of PSO. In vitro studies demonstrated that the nanoemulsion gel exhibited enhanced permeation and retention in the skin compared to commercial ointments, which were also confirmed through confocal laser scanning microscopy and dermatokinetic studies [125]. Table 5 presents the above-mentioned nanoemulsions.

### 6.5. Ethosomes (ESs)

Ethosomes are nanovesicles that are flexible and based on phospholipids with high levels (20–45%) of ethanol. Ethanol is recognized as an effective enhancer of permeation that is added to vesicular systems for creating flexible nanosheets. It has the ability to interact with the polar head group region of lipid molecules to reduce the lipid melting point of stratum corneum, thus promoting the increase of lipid fluidity and cell membrane permeability. The ethanol-derived vesicle membranes possess high elasticity, enabling the flexible vesicles to pass through pores smaller than their diameter. Ethosomal systems deliver substances to the skin more effectively in terms of the volume and depth compared to conventional liposomes or hydroalcoholic solutions [126]. 

The clinical efficacy of apremilast in oral therapy for psoriatic arthritis is limited by its low water solubility and permeability. Alfehaid and colleagues conducted research to enhance its efficacy by developing a nanoscale lipid carrier with a high payload and transdermal flux. The formulation displayed commendable physicochemical and rheological traits and was deemed appropriate for topical application. The gel also facilitated sustained drug release, while its high transdermal flux and brief lag time suggesting marked and prompt movement of the ethosomes across the skin barrier. The pharmacokinetic results demonstrated that ethosomes have potential in apremilast transdermal therapy, as the absorption rate was appropriate compared to the oral administration. In vivo results indicated that apremilast ethosomal gel, when administered transdermally, showed a higher bioavailability (225% relative bioavailability) than the orally administered suspension while maintaining a constant peak exposure. Briefly, the formulated ethosomal gel offers enhanced potential for transdermal therapy, making it suitable for once-daily application [127].

As was mentioned, acitretin is a second-generation retinoid used to treat severe PSO. However, its poor water solubility and systemic side effects limit its administration. The primary goal of Peram and colleagues’ study was to create and refine a nanogel containing acitretin-filled ethosomes and assess its potential for treating psoriasis locally. The results of the fluorescence microscopy study suggest that ethosomes have a greater ability to penetrate deep layers of the skin compared to the acitretin penetration rate from the gel without ethosome. The formulated ethosomal gel substantially increased the skin permeability and deposition during the ex vivo tests, while an in vivo study demonstrated a significant improvement in the therapeutic response [128].

In another study, Dadwal and colleagues created tacrolimus and hyaluronic acid ethosomal formulations using soy lecithin, ethanol and propylene glycol to overcome the problem that the regular use of tacrolimus can lead to serious side effects. The study found that the hyaluronic acid-based tacrolimus ethosome demonstrated sustained drug release, higher dermal flux, and a greater enhancement rate upon skin permeation. During the in vivo studies, the ethosome gel showed promise, and the hyaluronic acid had a synergistic effect when PSO was treated with tacrolimus [129]. Zhang et al. also demonstrated that hyaluronic acid could play an important role in the formulation and effectiveness of the ethosomal drug delivery systems. They covalently linked hyaluronic acid to propylene glycol-based ethosomes, resulting in a new drug delivery carrier suitable for the local application of curcumin. The findings demonstrated that the hyaluronic acid enhanced the release and the dermal delivery of poorly water-soluble curcumin. The amount of curcumin retained in the skin after 8 h, as well as the cumulative transdermal amount, increased by 1.6-fold in comparison with the ethosomes without hyaluronic acid. Furthermore, the in vivo curcumin retention in psoriatic skin with the hyaluronic acid containing ethosome was four times greater than that without the excipient. After topical application in mice, the group treated with ethosomes containing hyaluronic acid demonstrated a reduction in inflammatory symptoms, alongside decreased mRNA levels of TNF-α, IL-17A, IL-17F, IL-22 and IL-1β [130].

Resveratrol is a strong antioxidant and can be used to treat a variety of skin conditions. Therefore, Arora and Nanda’s research aimed to optimize the development of a transdermal delivery system for resveratrol to enhance its dermatological benefits. The study confirmed that the formulation variables could be appropriately manipulated to achieve the desired dermatological properties of the topical formulation, as supported by regression analysis. It was concluded that ethosomal hydrogel could be employed in the creation of topical formulations that offer improved dermatological benefits and are suitable and stable for patients’ use. However, more studies of skin irritation and cytotoxicity are required to assess and establish actual clinical acceptability [131].

Pleguezuelos-Villa and colleagues endeavored to integrate mangiferin, a naturally occurring compound extracted from *Mangifera indica*, into glycosomes, ethosomes, and glycerol-ethanol phospholipid vesicles (glycetosomes). In vitro research on human abdominal skin revealed the dose-dependent capacity of vesicles to encourage the retention of mangiferin in the epidermis. Additionally, glycetosomes demonstrated exceptional biocompatibility and demonstrated a potent ability to safeguard fibroblasts from peroxide-induced damage in vitro. The in vivo results verified the superior efficacy of glycetosome-encapsulated mangiferin in comparison to mangiferin dispersion in the promotion of wound healing, supporting their potential application in treating PSO [132].

Thymoquinone (TQ), a lipid-soluble benzoquinone, is the primary bioactive constituent found in the essential oil of *Nigella sativa*, known for its efficacy in treating psoriasis. However, due to its hydrophobicity, inadequate water solubility, and sensitivity to light, its development is limited. Thus, Negi et al. conducted a study to assess the anti-psoriatic potential of thymoquinone-filled ethosomal vesicles through a mouse tail model. They formulated a hydrogel, which was found to be rheologically acceptable, with substantial retention of thymoquinone in the dermal layers. It was found that the anti-psoriatic activity of the ethosomal hydrogel was significantly more pronounced compared with the marketed product [133]. The recently formulated ethosomes are summarized in Table 6.

### 6.6. Niosomes (NIOs)

Niosomes are vesicular drug delivery systems that rely on non-ionic surfactants. These systems offer multiple benefits, such as the capability to contain both hydrophilic and hydrophobic drugs, lower toxicity, enhance compatibility and promote biodegradation [134,135]. They are increasingly preferred over liposomes due to specific drawbacks, which include the chemical instability, high cost, and the vulnerability of phospholipids to oxidation [136]. The structure of niosomes limits the hydrophilic drug to the central aqueous compartment and the hydrophobic moiety to the bilayer matrix, proffering targeted, controlled, and sustained pharmaceutical delivery advantages [137]. Niosomes can be administered orally, parenterally, topically, and ocularly [138]. Non-ionic surfactants self-assemble to construct the framework of niosomes. The self-assembly of nonionic surfactants is influenced by thermodynamic effects, specifically entropy, and free-energy variation, as well as enthalpy contributions from forces like van der Waals, hydrogen bonding, hydrophobicity, and electrostatic interactions. Furthermore, the lipid chain length, aqueous interlayer, chain confinement, and membrane symmetry are among the geometrical factors that contribute to the self-assembly of nonionic surfactants, as observed by Marianecci et al. [139].

Considering the various drawbacks of current treatments for psoriasis, such as withdrawal rebound, high costs, and several adverse effects, Meng et al. formulated niosomes loaded with celastrol. Compared to the crude drug, celastrol niosomes exhibited enhanced in vitro permeability, while in vivo studies indicated that celastrol niosomes were effective in reducing redness and scaling on the back skin of mouse models with psoriasis. The study results indicate that the use of niosomes to encapsulate celastrol led to an increase in its water solubility and skin penetration, resulting in improved anti-inflammatory activity in mice [140].

Pandey et al. formulated a niosome gel for enhanced cyclosporine permeation and deposition to efficiently treat psoriasis locally. In ex vivo permeability studies conducted on rat skin, niosomes exhibited significant permeability rates (50.57%) after 24 h, surpassing that of a simple cyclosporine suspension (10.13%), which was used as a control [141].

Pentoxifylline is reported to reduce the side effects of cyclosporine and can also be used to treat psoriasis [142]. Considering this information, Bhardwaj et al. employed the thin-film hydration method to optimize niosomes filled with cyclosporine and pentoxifylline. The niosomes had a significant impact on the permeation of both drugs, as only a small amount of the drug was able to permeate through the skin and a substantial quantity of the drugs was retained in the stratum corneum. However, in vivo studies in mice with IMQ-induced psoriasis showed that niosomes loaded with pentoxifylline and cyclosporine significantly improved both the histological abnormalities and the Psoriasis Area and Severity Index compared to solutions containing each active ingredients separately [143]. Bhardwaj et al. conducted an in vitro and in vivo study to develop, optimize and test pentoxifylline niosomes in an IMQ-induced psoriasis model to enhance percutaneous delivery of the pentoxifylline. In this case, the niosomes containing pentoxifylline were constructed by the combination of cholesterol and Tween 80 (0.662 ratio), soy lecithin (14.5 mg) and pentoxifylline (10 mg). For studying the permeation and skin deposition of niosomes ex vivo, excised goat skin was used as the model. Notably, the ex vivo permeation studies did not detect any significant drug amount. However, the dermal deposition studies revealed that pentoxifylline niosomes exhibit deeper and more significant deposition than free drug in the epidermal layers [134]. In their other investigation, Bhardwaj et al. applied cholesterol and Span 60 to produce niosomes. According to the physical characteristics of the formulated niosomes, their most appropriate formulation was the combination of 1:2.2 of cholesterol and Span 60, 30 mg of cyclosporine, and a hydration time of 30 min. Ex vivo experiments were performed using excised goat skin. Although the drug permeation percentage was low, the amount of drug permeated through the skin from niosomes was considerably higher than that from the suspension. In the ex vivo permeation assays, niosomes exhibited a significant improvement in drug permeation through the goat skin in comparison to the suspension. The pathology and the PASI score confirmed that the skin condition of the mice treated with cyclosporine niosomes was significantly improved compared to those receiving the drug dispersion. Moreover, the niosomes maintained stability after undergoing storage stability tests for three months at both 40 °C and at room temperature. These results reveal the capability of non-ionic surfactant vesicles of cyclosporine to enhance the treatment of psoriasis while limiting the typical negative effects related to systemic delivery [144]. Table 7 presents the characteristics of the above-mentioned niosomes.

### 6.7. Nanosponges (NSs)

Nanosponges are hyper-crosslinked cyclodextrin polymers that generate nanostructured 3D networks by complexing cyclodextrin and crosslinker-like carbonyl diimidazole. Within their crystal architecture, several polymer chains can create distinct microdomains that are appropriate for entrapping drugs with differing chemical compositions. Nanosponges are noted for their effectiveness in dissolving drugs that have poor water solubility and facilitating prolonged release. They can load both hydrophilic and hydrophobic drug molecules, thanks to their internal hydrophobic cavities and external hydrophilic branches. Additionally, they have the potential to improve drug delivery in various therapeutic applications [145].

Curcumin, when combined with anti-inflammatory drugs like caffeine, displays an increased anti-patch effect compared to curcumin alone, thus reducing the time required to treat psoriasis. Considering this data, researchers aimed to formulate a nanosponge-based topical gel with a combination of curcumin and caffeine, which may serve as a potential therapy for PSO. The NS, which was comprised of dimethyl carbonate as a crosslinker and beta-cyclodextrin as a polymer, was created using the hot-melt method and mixed into topical gels formulated. The study found that the combination of curcumin and caffeine decreased the time required for detecting anti-psoriasis activity to 10 days as compared to curcumin alone, which took approximately 20 days. Furthermore, the nanogel provided a sustained release for 12 h [146].

In the therapy of psoriasis, clobetasol propionate, a topical corticosteroid, exhibits potential; however, the application of steroids to the skin causes detrimental effects such as skin atrophy, acne, hypopigmentation, and associated allergic contact dermatitis. Against these side effects, Kumar et al. developed a hydrogel containing nanosponges, which were synthesized using β-cyclodextrin and diphenyl carbonate. The formulation improved the solubility of pure clobetasol propionate in water by 45-fold, and the drug release value was found to be 86%. Additionally, in vitro cell viability assays were conducted using a human monocyte cell line (THP-1), which proved the biocompatibility of clobetasol propionate containing nanosponges. The results of the in vivo evaluation showed a remarkable reduction in the extent of orthokeratosis in comparison to the untreated group investigated in mice. Furthermore, the study revealed significant drug activity and a decrease in epidermal thickness [147]. 

As mentioned above, dithranol is widely regarded as the primary choice for treating psoriasis owing to its antioxidant, antiproliferative, and anti-inflammatory attributes. However, its poor stability and solubility significantly impede its effectiveness and dose determination. To tackle this issue, Kadian et al. conducted a comparative study on the effects of the type of preparation technique. They compared the stability, physicochemical characteristics, and efficacy of the dithranol-containing nanosponges created by a solvent evaporation technique or melt method. According to their study, the solvent evaporation technique proved to be a more efficacious approach for enhancing the stability and the solubility of the drug. Their findings affirmed that the application of the solvent evaporation technique to incorporate dithranol into nanosponges not only resulted in amplified solubility and photostability but also sustained the antioxidant efficacy of the selected medication [148]. The objective of another study was to formulate and evaluate dithranol nanosponges integrated into Carbopol hydrogel. The application of these nanosponges containing dithranol (0.5 and 1.0% *w*/*v*) displayed a significant increase in the degree of epidermal thickness when compared to the untreated control group. This research presented a prospective model of a multifunctional cyclodextrin nanosponge hydrogel to facilitate the establishment of a promising topical treatment method for psoriasis, thus broadening current treatment alternatives while avoiding the potential for systemic adverse side effects. Ongoing advances in the field of new therapeutics may be implemented in commercial dosage forms shortly [149]. Table 8 is intended to present the above-mentioned nanosponges and their parameters.

### 6.8. Dendrimers (Ds)

Dendrimers are hyperbranched macromolecules with a three-dimensional structure, as their terminal functional groups may chemically bind to other molecules, altering the surface properties for purposes such as nano-device biomimetics. They have a central core encircled by peripheral groups, while the diameter of dendritic macromolecules is directly proportional to their formation, resulting in a spherical shape [145]. Dendrimers have become promising delivery carriers for investigating the impact of the polymer size, charge, and composition on biologically significant properties, including interactions with lipid bilayers, internalization, plasma retention time, biodistribution, and filtration [150]. 

Despite the listed advantages, not many external preparations containing dendrimers can be found in the literature. Some research studies have explored the use of dendrimers in delivering anti-inflammatory drugs or other medications commonly used to treat psoriasis, such as dithranol, isotretinoin, corticosteroids or calcineurin inhibitors. By encapsulating these drugs within dendrimers, researchers aim to improve their penetration into the skin and enhance their therapeutic effects. 

Dendrimers can be synthesized through a stepwise process known as divergent or convergent synthesis, beginning the preparation with a core molecule or starting with multiple branches, respectively. The choice of the technique depends on factors such as the desired dendrimer structure, functionality, and intended applications [151]. Additionally, various types of dendrimers, including polyamide amine (PAMAM) and polypropylene imine (PPI), can be synthesized using these methods, each with its own specific synthetic protocols. For example, Tripathi et al. evaluated the use of PAMAM dendrimers in administering dithranol through a unique microsponge-centered gel. This formulation was determined to be stable and non-irritating when administered to the skin of test animals. Additionally, the pharmacokinetic findings indicated that the microsponge formulation of dithranol, encapsulated in PAMAM, can prolong the skin penetration equivalent of the drug compared to the marketed form of dithranol. The formulated nanocomplex increased the solubility of dithranol and enhanced skin permeability, while it reduced irritation and increased patient compliance, with improved physical, chemical, and thermal stability [152]. Agrawal et al. investigated the capacity of PPI dendrimers prepared by a divergent method (ethylenediamine was taken as an initiator core and acrylonitrile was added to it) to transport dithranol for topical treatment. Their formulation achieved a notable enhancement in drug permeation, increasing from 35% to 95%, and a reduction in skin irritation compared to a dithranol solution [153].

A novel administration method for isotretinoin was reported by Zhao et al., aiming to improve the efficacy of this retinoid in treating dermatological ailments locally. Although isotretinoin is not a first-line treatment for PSO, there are some studies suggesting that it may have some efficacy in certain cases, particularly pustular PSO. The scientists developed a self-assembling dendrimer-conjugated system for transdermal administration of isotretinoin. Remarkably, it exhibited a significant and controlled release profile, gradually discharging in regular tissues and accelerating the release in acidic environments such as inflamed sites. These advantageous release characteristics could reduce the teratogenic side effects of isotretinoin, facilitating effective skin penetration. In vitro and in vivo studies showed that dendrimers improved the penetration rate of the isotretinoin to the lower layers of the skin, thus targeting lesions more effectively [154]. Table 9 summarizes the above-mentioned dendrimers.

### 6.9. Nanocrystals (NCs)

Nanocrystals (NCs) exhibit high drug-loading capacity, low toxicity, and strong structural stability, which renders them a promising and adaptable approach for boosting local delivery of insoluble drugs. This is accomplished by augmenting the skin adhesion, concentration gradients, and hair follicle accumulation. The formulation development to promote the passive diffusion and/or follicular targeting of nanocrystals holds significant importance within clinical practice [155].

18β-glycyrrhetinic acid is frequently applied topically to address inflammatory skin conditions in clinical practice. Nevertheless, its water insolubility leads to decreased bioavailability and limited skin permeability. In line with the hypothesis that the permeability and bioavailability of the drugs can be enhanced through nanocrystallization, Quan and his co-workers developed 18β-glycyrrhetinic acid nanocrystals using high-pressure homogenization. Their formulation suppressed the expression of proinflammatory cytokines on tetradecanoyl phorbol acetate (TPA)-induced edema in mouse ears, reduced the myeloperoxidase activity and neutrophil infiltration, and exhibited a potent anti-inflammatory activity [156].

Xiang et al. examined the collective impact of the particle size, penetration enhancers, and transport enhancers on the build-up of nanocrystals, as well as their passive permeation through the skin. They created a composite of curcumin nanocrystals, propylene glycol, and xanthan gum gel. The mixture was then tested on porcine skin. The results demonstrated that xanthan gum decreased the penetration of curcumin nanocrystals into the follicles and reduced their accumulation in the skin. Propylene glycol enhanced the permeation and retention of curcumin nanocrystals in vitro for 24 h. Curcumin nanocrystals with particle sizes of 60 and 120 nanometers exhibited superior passive permeation through the skin compared to those with a size of 480 nm, but they also demonstrated more profound follicular accumulation [155].

Several investigations are available on the effectiveness of incorporating different anti-inflammatory plant active ingredients into nanocrystals. For example, an indirubin-a naturally occurring chemical compound with anti-inflammatory effect-containing NCs was created by Li et al. and incorporated into a hydrogel base. They used hyaluronic acid to target the CD44 cell-surface glycoprotein, which is overexpressed in inflamed psoriatic skin. The effectiveness of their formulation was proved in an IMQ-induced psoriasis model in mice, resulting in the relief of psoriasis symptoms and a decrease in the corresponding cytokine levels. The indirubin NCs encapsulated in the hyaluronic acid hydrogel enhanced the absorption of indirubin. It was observed that indirubin containing NCs specifically adhered to CD44, resulting in an increase in indirubin accumulation in the inflamed skin. Furthermore, the formulation augmented the anti-psoriatic impact of indirubin in both mouse models and HaCaT cells [157]. Diosmin, another natural active agent (flavonoid), also displays anti-inflammatory and antioxidant characteristics. Nevertheless, its physicochemical attributes pose difficulties since its solubility demands a pH of 12, consequently impacting its bioavailability. The study conducted by Shahine et al. therefore aimed to establish and characterize antisolvent precipitation techniques for the development and production of diosmin NCs. An in vivo evaluation was conducted to assess and compare the efficacy of diosmin nanocrystal gel and diosmin powder gel at three different doses in managing IMQ-induced psoriasis in rats. The study showed a significant reduction in the PASI score and serum inflammatory cytokine levels, and decreased expression of NF-κB expression in psoriatic skin tissue [158]. Rutin is also a polyphenolic flavonoid with a broad therapeutic spectrum, but with low water solubility and limited bioavailability. For these reasons, Hassan and colleagues prepared rutin nanocrystals by adopting an antisolvent nanoprecipitation-ultrasonic treatment technique. According to their results, nanocrystals stabilized by hydroxypropyl β-cyclodextrin (HP-β-CD) presented the smallest particle size, the highest drug entrapment efficiency, the greatest colloidal stability, as well as the highest drug photostability. The solubility of the drug was raised by a consecutive 102–202-fold, with the nanocrystals acting as stabilizers, while the dissolution rate was amplified by 2.3–6.7 times. HP-β-CD nanocrystalline hydrogels, in comparison to free-drug hydrogels, exhibited a higher rate of drug release and penetration into mouse skin. In vivo studies demonstrated that these rutin nanocrystalline hydrogels had a notably superior edema reduction effect when compared to free-agent hydrogels or commercial diclofenac sodium gels [159]. So and colleagues fabricated cellulose nanocrystals from *Gelidium amansii* (a species of red algae) and evaluated their anti-inflammatory properties on human keratinocytes and mouse skin. The NCs showed no cytotoxic effects on HaCaT cells and the downregulation of the proinflammatory factors was also proved. The treatment with the algae NCs significantly suppressed the increase in epidermal thickness and cyclooxygenase-2 (COX-2) expression in mouse skin induced by acute UVB radiation [160]. Table 10 lists the above-mentioned nanocrystals and their parameters.

### 6.10. Polyelectrolyte Nanoparticles

Polyelectrolyte nanoparticles have also shown promise in topical treatments for psoriasis due to their unique properties and potential therapeutic effects. These particles are composed of polymers that contain ionizable groups, resulting in a charged structure when dispersed in a suitable solvent. They can be designed to carry drugs or therapeutic agents due to their ability to interact with oppositely charged molecules, forming stable complexes. They can improve the efficacy of drugs used in psoriasis treatment by protecting them from degradation, enhancing their solubility, and prolonging their release. This can lead to more sustained therapeutic effects and potentially reduce the frequency of application required for treatment [161,162]. 

Polyelectrolyte complexes (PECs) based on chitosan and fucoidan garner significant interest in biomedical applications due to their unique properties and potential therapeutic benefits. These complexes are formed through electrostatic interactions between oppositely charged polyelectrolytes. Chitosan is positively charged due to the presence of amino groups, while fucoidan is negatively charged due to sulfate groups. Fucoidan is a sulfated polysaccharide found in various species of brown seaweed. It exhibits various biological activities, including anti-inflammatory, antioxidant, anticoagulant, and anticancer properties. It has been investigated for its potential therapeutic applications in wound healing, cancer therapy, and inflammatory diseases. The combination of chitosan and fucoidan in polyelectrolyte complexes can lead to synergistic effects, enhancing the therapeutic efficacy of the encapsulated drugs or bioactive compounds. Additionally, the mucoadhesive properties of chitosan can improve the residence time of PECs on mucosal surfaces, enhancing drug absorption and bioavailability [162].

Barbosa and colleagues developed PECs utilizing fucoidan and chitosan to enhance skin permeability for localized delivery of methotrexate. These polymeric particles exhibited a loading efficiency of approximately 14% and encapsulation efficiency ranging from 80% to 96%. These complexes had a size between 300 and 500 nm, with zeta potential values displaying both positive (+60 mV) and negative (−40 and −45 mV) charges, depending on the mass ratio of fucoidan to chitosan. Encapsulation of methotrexate resulted in reduced cytotoxicity compared to the free form when tested against fibroblasts and HaCaT cell lines. Skin permeability studies conducted using a porcine ear skin model revealed that the permeability of negatively charged particles was 2.7- and 3.3-fold higher compared to free methotrexate [163].

### 6.11. Polymersomes

Polymersomes are synthetic vesicles that enclose an aqueous cavity. They are formed through the self-assembly of amphiphilic copolymers [164,165]. Polymersomes are versatile and stable systems with adjustable properties, including drug encapsulation and release capabilities. These properties can be easily modified using various biodegradable and stimuli-responsive block copolymers. Polymersomes are widely regarded as one of the most promising supramolecular structures for delivering drugs, genes, and proteins in the fields of nanomedicine and nanobiology due to their numerous advantages [166].

Marepally and colleagues developed a new system called fuming nucleic acid-lipid particles (F-NALPs) that contained two therapeutic nucleic acids: anti-STAT3 siRNA (siSTAT3) and anti-TNF-a siRNA (siTNF-a). The nanocarrier system employed a newly synthesized cationic amphiphilic lipid with oleyl chains. The F-NALPs had a hydrodynamic size of 102 ± 6 nm and a surface potential of 32.14 ± 6.21 mV. The delivery of fluorescein isothiocyanate siRNA by F-NALPs reached a depth of 360 µm into the skin. Immunoblot analysis showed that treatment with double siRNA polymerosomes decreased the expression of STAT3, TNF-α, NF-κB and IL-23 compared to the free double siRNA solution. The efficacy was evaluated using an IMQ-induced psoriasis model in mice after 5 days of daily treatment. The study found that the use of commercially available tacrolimus ointment (Topgraf^®^) in combination with double siRNA polymerosomes resulted in lower PASI scores compared to the use of free double siRNA solution, STAT3 siRNA polymerosomes, and TNF-α siRNA polymerosomes alone [167]. 

### 6.12. Transfersomes

The supramolecularly structured aggregates of transfersomes make them highly deformable and able to pass through small pores. They have a similar morphology to liposomes but are functionally deformable enough to pass through pores much smaller than their own size. Transfersomes are vesicular particles composed of ultradeformable lipid bilayers and at least one inner aqueous compartment [168].

Bhatia et al. developed tamoxifen-loaded transfersomes using phospholipids and Span 80 as a surfactant. These were then incorporated into a Carbopol hydrogel. The anti-psoriatic efficacy of the tamoxifen-transfersome-loaded gel was assessed by comparing orthokeratosis in a murine tail model. After 4 weeks of daily treatment, the tamoxifen-transfersome-loaded gel led to significantly higher orthokeratosis than a free tamoxifen gel. Phospholipid-rich transfersomes were able to interact favorably with skin lipids, which could explain their effectiveness [169].

Todke and colleagues developed transfersomes that combined clobetasol propionate and cyclosporine. The transferosomes were produced by the thin-layer hydration method. They prepared nanoscale transferosomes (<150 nm) with high cyclosporine encapsulation efficiency (>86%). Ex vivo results demonstrated that these transfersomes could effectively transport cyclosporine and clobetasol propionate to the skin. RT-PCR assays indicated that the optimized formulation reduced the levels of TNF-α and IL-1 [170].

Gizaway and colleagues successfully developed and optimized a formulation of betamethasone dipropionate-loaded transfersomes. The formulation demonstrated excellent stability and exhibited spherical, single-cell vesicles. The formulation demonstrated good stability at both 4 °C and 25 °C for six months. The results indicated significant clinical improvement and a notable increase in safety and tolerability with the transfersomes containing gel compared to the cream containing free betamethasone dipropionate [171].

Parkash and colleagues developed transfersomes containing tacrolimus using a rotary evaporation method. Their research demonstrated that tacrolimus had superior permeability when it was encapsulated within transfersomes compared to conventional liposomes. The enhanced permeability was evaluated using various pharmacokinetic and pharmacodynamic parameters, which suggested the improved absorption and distribution of tacrolimus within the skin [172]. 

### 6.13. Emulsomes

Emulsomes are a superior type of lipid-based colloidal delivery system that combines the best features of both liposomes and emulsions. They consist of a phospholipid bilayer surrounding an oil core, similar to liposomes, but with the addition of a surfactant to stabilize the lipid bilayer. This unique and innovative structure provides emulsomes with several advantages, including significantly increased stability, enhanced drug-loading capacity, precisely controlled release, and vastly improved bioavailability of encapsulated drugs. Emulsomes are an excellent choice for delivering hydrophobic drugs due to their lipid core, which provides a suitable environment for solubilization [173].

Raza et al. developed a novel formulation using emulsomes containing dithranol, which were incorporated into a Carbopol^®^ hydrogel for ease of application. The study aimed to evaluate the anti-psoriatic activity of this formulation compared to a commercially available dithranol ointment (Derobin^®^). The results showed that the dithranol-emulsome-loaded gel induced higher orthokeratosis (52%) compared to Derobin^®^ ointment (41%) after 3 weeks of daily treatment. This research clearly showed that the dithranol-emulsome-loaded gel was significantly more effective than Derobin^®^ while avoiding the adverse effect of redness associated with Derobin^®^ ointment. Compared to the commercial ointment, the study demonstrated that the novel gel formulation had a significantly milder effect on the skin. Furthermore, the results indicated that a higher phospholipid content was directly correlated with an increase in dithranol retention, which undoubtedly contributed to the superior therapeutic efficacy of the gel formulation [174]. 

Based on the data presented by Gupta et al., emulsomes were the most effective of the compared vesicular systems in delivering the drug through the skin. Emulsomal gels were safe when applied topically and did not cause irritation. Moreover, the emulsomal gel significantly improved the accumulation of the active ingredient in the skin layers compared to the plain medicated gel. These results highlighted the efficiency of emulsomes in enhancing the local delivery of the drug to the affected areas of the skin, providing a valuable advantage in treating psoriasis [175]. 

### 6.14. Gold Nanoparticles 

Gold nanoparticles (AuNPs) have unique properties that make them ideal for a wide range of medical and non-medical applications. They are biocompatible, inert, and have low toxicity. AuNPs can be produced on a large scale using various physical, chemical, and biological methods as a reducing agent under different conditions, making them a versatile and valuable resource. Microbes can be used to produce gold nanoparticles both internally and externally, but external treatment simplifies the isolation process. AuNPs are an exceptional tool in diagnostic and therapeutic applications, such as biosensors, targeted cancer therapy, and precise drug delivery to diseased tissues. Their unique properties make them a promising tool in the field of nanomedicine [176]. 

Nemati et al. effectively treated psoriasis by conjugating siRNAs targeting the epidermal growth factor receptor (EGFR) to gold nanoparticles. The study conclusively showed that EGFR siRNA-AuNPs significantly reduced the EGFR expression compared to nonsense siRNA-AuNPs. Immunohistochemistry staining showed a significant decrease in the expression of T cell markers CD3, CD4, and CD8 with EGFR siRNA-AuNPs compared to nonsense siRNA-AuNPs. The efficacy of these therapies mixed with Aquaphor^®^ was tested in an IMQ-induced psoriasis murine model after treatment three times per week for 21 days. Application of these AuNPs resulted in a significant reduction in epidermal thickness, while no significant change was observed with free EGFR siRNA. These findings suggested that topical application of the mentioned AuNPs could effectively improve psoriasis by modifying gene expression and reducing T cell production [177]. 

The study performed by Fereig et al. also demonstrated that gold nanoparticles conjugated with tacrolimus-containing lecithin–chitosan hybrid nanoparticles were significantly more effective in treating psoriasis than bare hybrid nanoparticles without gold. The in vivo study results provided promising evidence of the anti-psoriasis effect when using gold-conjugated tacrolimus-containing lecithin–chitosan hybrid nanoparticles. Compared to bare hybrid nanoparticles, our study found significant differences in some inflammatory markers. Furthermore, these gold conjugates demonstrated an anti-inflammatory effect, as evidenced by a lower spleen/body weight ratio and better histopathological skin condition compared to other preparations investigated [178]. 

The research conducted by Özcan et al. provided clear evidence that the combination of MTX and gold nanoparticles was more effective in reducing inflammation than MTX alone. The combined treatment resulted in a significant reduction in T cells, CD4+ T cells, and neutrophils. Additionally, it was well-tolerated both systemically and locally. It was concluded that MTX-AuNPs significantly impacted the immune and stromal components of the skin, effectively inhibiting pathogenesis in preclinical PSO [179]. 

### 6.15. Solid Polymeric Nanoparticles

Polymeric nanoparticles are indeed versatile carriers of various active compounds, including drugs, imaging agents, and other bioactive molecules. They are characterized by their nanoscale size, typically ranging from 1 to 1000 nanometers, and can be loaded with active compounds either entrapped within the polymeric core or surface-adsorbed onto the polymer surface [180]. 

Asad et al. successfully prepared a chitosan hydrogel loaded with MTX-based polymeric nanoparticles. The optimized MTX-NPs demonstrated a particle size of 256.4 ± 2.17 nm and an impressive encapsulation efficiency of 86 ± 0.03%. The ex vivo permeation test revealed that only 19.95 ± 1.04 µg/cm^2^ of the active substance permeated through the skin in 24 h, while the epidermis retained 81.33% of the active substance. The study demonstrated a significant decrease in the PASI score, indicating a remarkable improvement in the normal skin condition of the mice. The MTX-NPs hydrogel caused minimal hyperkeratosis and parakeratosis. Histopathological examinations indicated skin healing in mice [181]. 

Mao and his colleagues developed a new amphiphilic polymer to aid in the delivery of curcumin to deeper layers of the skin. Silk fibroin was used as a hydrogel-based matrix to further facilitate local delivery of the model drug. They demonstrated that the silk fibroin gel, which contained cationic nanoparticles, improved the skin permeation of curcumin. In vitro studies showed that the addition of curcumin NPs to the hydrogel led to a slower release of the active substance compared to plain curcumin gel. In vivo studies indicated that the curcumin-NPs gel was more effective in inhibiting the expression of inflammatory cytokines such as TNF-α, NF-κB, and IL-6 [182]. 

Chamcheu et al. encapsulated (−)-epigallocatechin-3-gallate (EGCG) in chitosan-based solid polymeric nanoparticles. After two weeks of daily treatment, the use of these NPs improved erythema, scaling, and thickness compared to free EGCG. The effects of nanoEGCG were equal to or better than those obtained with over 20 times higher doses of free EGCG. The results of the immunohistochemistry analysis indicated that treatment with the polymeric nanoparticles led to a greater reduction in the levels of mast cells, neutrophils, macrophages, and CD4+ T cells compared to free EGCG. The enhanced anti-proliferative, anti-inflammatory, and pro-differentiation effects of nanoEGCG indicated that it retained the mechanistic activity of free EGCG while improving its delivery and effectiveness [183].

### 6.16. Nanogels

Nanogels are cross-linked polymer nanoparticles that serve as highly efficient and biodegradable carriers for controlled drug delivery. They contain pores that enable the high encapsulation of bioactive compounds or drugs due to the specific charge of the polymers. These hydrogel particles are only a few nanometers in size, exhibiting both hydrogel and nanoparticle properties. Nanogels can be prepared from polymeric precursors or by heterogeneous polymerization of monomers. Physical or chemical cross-linking is a crucial step in their preparation [184]. 

Nirmal and colleagues developed a dual chitosan nanogel loaded with 2,2’-azobis [2-(2-imidazolin-2-yl)propane]-dihydrochloride (AIPH) and 8-hydroxypyrene-1,3,6-trisulfonic acid (HPTS). The in vivo topical nanogel treatment induced a potent necrosis effect, which alleviated the epidermal thickening and inflammation in an IMQ-induced animal model. The experimental data in this study indicated that nanogels were effective carriers for the local permeation of alkyl radicals in psoriasis. They could deliver the necessary drug to induce cell death into the deeper epidermis without causing skin irritation [185].

Panonnummal and Sabitha investigated the anti-psoriatic activity of MTX-loaded chitin nanogels (MCNG), both in vitro and in vivo, to enhance the penetration of MTX. The particles were spherical, with a size of 196 ± 14 nm and a surface charge of +9.21 ± 0.42 mV. Skin permeation studies showed an increased transdermal flux of methotrexate from MCNG in the stratum corneum and other epidermal skin layers compared to samples treated with a control MTX solution. After three weeks of daily use, the nanogels led to significant reductions in the PASI scores when compared to a free methotrexate gel. Their study showed increased skin permeation and retention with the nanogels compared to the free methotrexate gel. The positively charged, hydrophobic nanogels likely increased the interaction with anionic skin lipids and hydrophobic skin keratinocytes, improving MTX retention and increasing efficacy at lower doses than free MTX [186]. In another study, chitosan/hyaluronan nanogels were developed to co-deliver MTX and 5-aminolevulinic acid (ALA). The nanogels improved the penetration and retention of MTX and ALA through the skin, both in vitro and in vivo, compared to the MTX-ALA suspension. Additionally, the nanogels enhanced cellular uptake and decreased reactive oxygen species generation. The study found that the combination of MTX-ALA NGs resulted in a synergistic anti-proliferation effect and apoptosis in lipopolysaccharide-irritated HaCaT cells. In vivo, the MTX-ALA NGs were effective in improving skin manifestations and reducing proinflammatory cytokines TNF-α and IL-17A. Additionally, the NGs increased the skin permeability of both MTX and ALA, thus enhancing their therapeutic effectiveness against IMQ-induced psoriatic-like plaques in BALB/c mice [187].

## 7. Clinical Studies on Nanocarriers Containing Anti-Psoriatic Drugs

Only a few clinical studies have been conducted to evaluate the effectiveness of various innovative nanocarriers developed through novel techniques for the treatment of PSO. While some of these studies have demonstrated encouraging outcomes, others have not yielded convincing results. Table 11 summarizes the latest clinical trials on nanocarriers containing anti-psoriatics.

In a study conducted by Nasr et al., a jojoba oil-based nanoemulsion of tazarotene (0.1% *w*/*w*) was topically applied for psoriasis treatment. This emulsion exhibited a 7.6-fold greater reduction in the PASI scores of psoriatic patients compared to the commercially available gel. Additionally, the emulsion with a particle size of 15.5 nm showed no skin irritation, unlike the marketed gel, which caused skin redness and inflammation [188]. A jojoba oil-based nanoemulsion was also investigated in another study conducted by Ramez et al. In this case, methotrexate was incorporated into the nanocarrier containing 40% jojoba oil, 45% Tween-80 and Span-85 at a ratio of 3:1, and 15% water. The effectiveness of the formulation was investigated in 30 patients with plaque psoriasis for 8 weeks. A statistically significant reduction in the severity (TES (thickness, erythema, scales)) score for the psoriatic plaque was found. This clinical study demonstrated that MTX nanoemulsion prepared from jojoba oil was both safe and clinically promising in the treatment of PSO [189].

In another recent clinical study, involving 20 patients, the effectiveness of a nanoemulsion containing a natural active ingredient with antioxidant and anti-inflammatory properties (*Rosa damascena*) has been described. The nanoemulsion was formulated with the help of isopropyl myristate, Cremophor RH 40 and Transcutol HP, and it contained 5% *Rosa damascena* extract. At the end of the treatment (6 weeks), an improvement in the quality of life of the patients was observed and the erythema induration was reduced [190].

Curcumin-containing niosomes were formulated by Kolahdooz et al. to improve the delivery and anti-inflammatory ability. The team prepared niosomes using the thin-film hydration technique and added them to a hyaluronic acid and marine collagen gel-based formulation. Five patients with mild-to-moderate psoriasis, aged between 18 and 60 years old, and with PASI scores below 30, exhibiting symmetrical and comparable lesions, were enrolled in this study. The formulation containing 15 µM curcumin was topically administered to the skin lesions for a duration of 4 weeks and compared against a placebo. The treatment resulted in a significant reduction in redness and exfoliation and an overall improvement compared to the placebo-treated group. Gene expression analyses revealed noteworthy reductions in IL-17, IL-23, IL-22 and TNF-α in lesions treated with the niosome-containing gel, suggesting its therapeutic potential for patients with mild-to-moderate PSO by suppressing the immunopathogenic axis of IL-17/IL-23. In summary, curcumin-containing niosomes may provide an effective therapeutic approach for this cohort of patients. The niosomal gel could potentially enhance the therapeutic outcome of PSO skin lesions and promote the healing process as a topical treatment [191].

In a randomized clinical trial conducted by Fathalla et al., liposomal and ethosomal gels containing dithranol have been investigated in 20 psoriatic patients. After treatment, the PASI score decreased by 68.66% and 81.84% for liposomes and ethosomes, respectively. According to their results, dithranol ethosomes were much more effective than liposomes [192]. In an earlier study, a liposomal formulation of dithranol was synthesized, and a clinical trial with 20 patients suffering from plaque PSO was conducted for 6 weeks. Their formulation showed low irritation and minimal staining of the skin and clothes. They also found the efficacy of the liposomal gel was the same as the cream without nanoparticles [193].

In another randomized clinical trial, Kumar et al. investigated the efficacy and safety of topical cyclosporine-loaded liposomal gels composed of high-purity phosphatidylcholine. Thirty-eight patients with mild-to-moderate chronic plaque PSO were selected for the study. It was found that 95% of the patients treated with the liposomal gel containing 2% cyclosporine responded positively to the treatment according to the physician rating. Moreover, it was observed that there was no accumulation of cyclosporine in the systemic pool in any of the patients, even with repeated applications. However, adverse reactions, such as erythema, irritation, and dryness, were noted in 8.8% where cyclosporine lipogel was applied, and one patient reported skin cracking at the treated sites [194].

**Table 11 pharmaceutics-16-00449-t011:** Recent clinical studies using nanoparticles for the treatment of psoriasis.

Carrier Type	Anti-Psoriatic Drugs	Number of Patients	Clinical Outcome	Ref.
Liposome	Cyclosporine	38	Improved effectiveness; appearance of adverse reactions	[194]
	Dithranol	20	Lower irritation and minimal staining of skin and clothes compared to the commercial cream. Same efficacy as the cream without nanoparticles.	[193]
Liposome and Ethosome	Dithranol	20	Decreased PASI score	[192]
Nanoemulsion	Tazarotene	20	7.6-fold greater reduction in the PASI score of psoriatic patients compared to the commercially available gel	[188]
	Methotrexate	30	Reduced psoriatic plaques without causing skin irritation	[189]
	*Rosa damascena*	20	Improvement in the quality of life of the patients; reduced erythema induration	[190]
Niosome	Curcumin	5	Significant reduction in redness and scaling; significant downregulation of proinflammatory cytokines	[191]

PASI, Psoriasis Area and Severity Index.

## 8. Nanoparticle Selection According to the Type of Psoriasis

Nanoparticles hold promise in the treatment of various types of PSO due to their ability to deliver drugs directly to affected areas, enhancing efficacy and reducing side effects. The physico-chemical properties of the above-mentioned NPs can determine which type of PSO they can be effective in [195]. For plaque psoriasis, nanoparticles such as liposomes or polymeric nanoparticles can be beneficial, which is why researchers extensively employed these types of drug carriers for targeted topical treatment of psoriatic plaques. Liposomes can encapsulate drugs and penetrate the thickened skin layers typical of plaque psoriasis, delivering medications directly to the affected areas [9]. Polymeric nanoparticles offer controlled release of the active substances, ensuring a prolonged therapeutic effect and minimizing the need for frequent application [196]. Nanogels or dendrimers could be effective in guttate psoriasis because they provide sustained release, suitable for treating the dispersed lesions characteristic of guttate PSO [197]. Dendrimers also have potential anti-inflammatory properties, making them suitable for targeting guttate psoriasis lesions [198]. For the pustular type, lipid-based nanoparticles like SLNs or NLCs may be beneficial, as they can encapsulate both hydrophilic and lipophilic drugs, offering flexibility in drug delivery. Additionally, their lipid composition can provide moisturizing effects, which are beneficial for managing the dry, inflamed skin associated with pustular psoriasis [95,197]. Nanoemulsions could be effective delivery systems in the case of the inverse PSO, as they can offer enhanced skin penetration and drug solubilization, making them suitable for targeting the folds and creases [115,122].

## 9. Conclusions and Future Perspectives

Psoriasis, a persistent autoimmune skin disorder affecting people globally, poses a significant treatment challenge. While various strategies are employed to address its severity, finding a safe, effective, and complete cure remains elusive. Topical therapy, due to the reduced toxicity and improved patient compliance, remains a primary treatment option. However, traditional topical agents face limitations such as poor drug penetration and dose-related toxicity, necessitating innovative approaches for treating this challenging skin condition. Novel drug delivery systems offer a promising solution by overcoming issues associated with conventional methods. Over the past century, one of the most remarkable steps in science has been the appearance of nanotechnology, which holds promise as a potential solution for controlling skin disorders, as well. Nanocarriers, like liposomes, solid lipid nanoparticles (SLNs), nanostructured lipid carriers (NLCs), nanoemulsions, ethosomes, dendrimers and nanocrystals, have become increasingly popular for delivering drugs topically in the treatment of psoriasis, and they hold promise for addressing some of the challenges associated with conventional drug delivery methods. The challenge when treating dermatologic conditions, including psoriasis, atopic dermatitis, and acne, is to achieve optimal epidermal penetration and retention while minimizing drug absorption into the bloodstream to avoid serious side effects. Psoriasis presents an additional hurdle due to the thickening of the stratum corneum, making it a formidable barrier for active ingredients. Numerous studies have explored this challenge, with a focus on both polymeric and lipidic nanoparticles, over the past few decades. It was described by many research groups that nanocarriers are able to enhance the drug penetration rate, can be designed to release drugs in a controlled and sustained manner, and can be engineered to target specific cells or tissues, allowing for precise drug delivery. However, despite these advantages, nanoparticles are still in early development, facing challenges like high costs, long-term stability concerns, and potential toxicity. To address these issues and ensure effective clinical translation, rigorous preclinical models are necessary to understand the interactions of these nanocarriers with the skin.

In conclusion, the key objectives for effectively managing psoriatic plaques involve achieving improved skin penetration and deposition while minimizing systemic absorption. Employing well-suited nanocarriers in combination with the right active ingredients holds the potential to attain this goal. The formulation methods and results discussed in this review serve as a valuable foundation for future research, providing insights that may aid other researchers in exploring and refining these approaches using different nanoparticles and anti-psoriatic drugs together. These approaches could transform psoriasis into a manageable condition, exerting minimal physiological and psychological impacts on affected individuals.

## Figures and Tables

**Figure 1 pharmaceutics-16-00449-f001:**
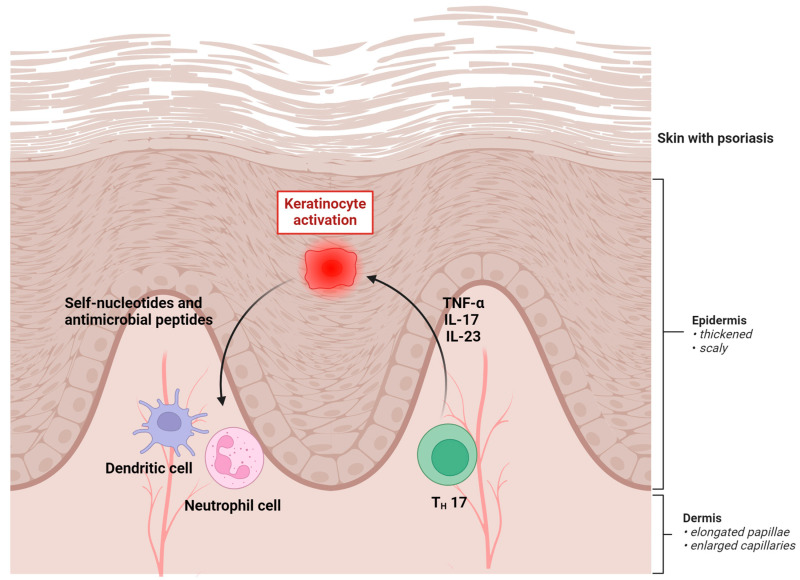
A schematic illustration of the pathomechanism of psoriasis.

**Figure 2 pharmaceutics-16-00449-f002:**
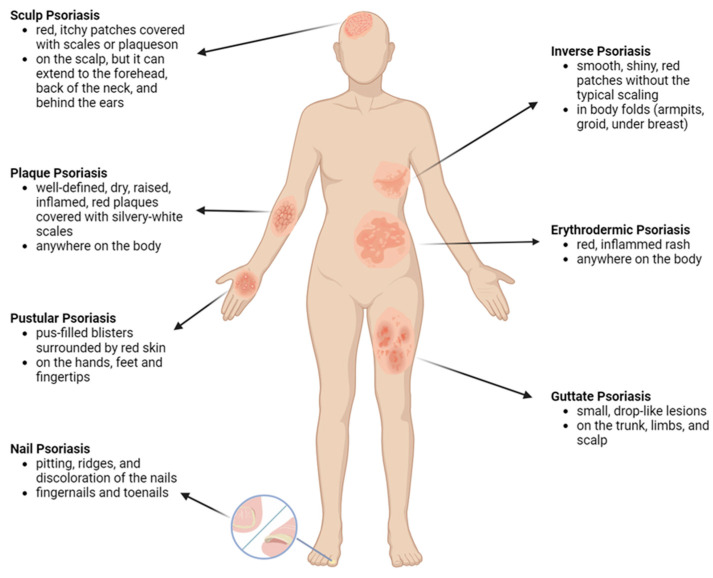
The main types of psoriasis.

**Figure 3 pharmaceutics-16-00449-f003:**
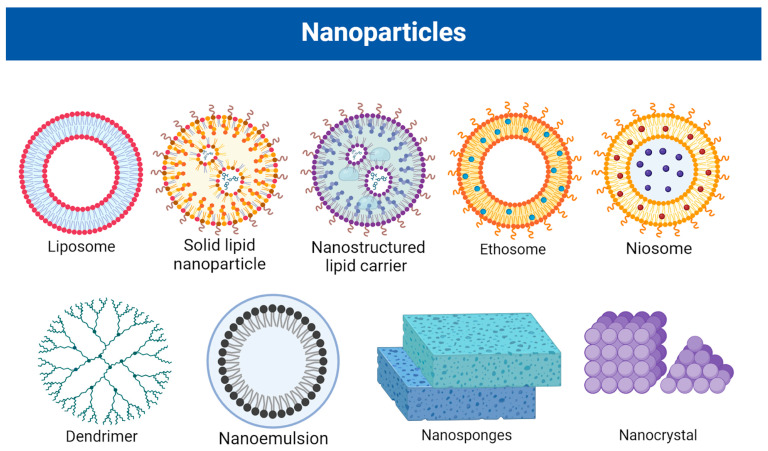
The most commonly used nanoparticle types (created with BioRender.com).

**Table 1 pharmaceutics-16-00449-t001:** Classification of corticosteroids.

Potency Group	Corticosteroid	Vehicle Type
Super-high potency (Class 1)	Betamethasone dipropionate	Ointment, Gel, Lotion
	Clobetasol propionate	Cream, Gel, Lotion, Shampoo, Foam, Aerosol, Ointment, Solution
	Diflucortolone valerate	Ointment, Oily Cream
	Fluocinonide	Cream
	Flurandrenolide	Tape (Roll)
	Halobetasol propionate	Lotion, Cream, Ointment, Foam
High potency (Class 2)	Amcinonide	Ointment
	Betamethasone dipropionate	Ointment, Cream
	Clobetasol propionate	Cream
	Desoximetasone	Cream, Ointment, Gel
	Diflorasone diacetate	Ointment, Cream
	Fluocinonide	Gel, Cream, Solution, Ointment
	Halcinonide	Ointment, Solution, Cream
	Halobetasol propionate	Lotion
High potency (Class 3)	Amcinonide	Cream, Lotion
	Betamethasone dipropionate	Cream
	Betamethasone valerate	Ointment, Foam
	Desoximetasone	Ointment, Cream
	Diflorasone diacetate	Cream
	Diflucortolone valerate	Ointment, Cream
	Fluocinonide	Cream
	Fluticasone propionate	Ointment
	Mometasone furoate	Ointment
	Triamcinolone acetonide	Ointment, Cream
Medium potency (Class 4)	Betamethasone dipropionate	Spray
	Clocortolone pivalate	Cream
	Fluocinolone acetonide	Ointment
	Flurandrenolide	Ointment
	Fluticasone propionate	Cream
	Hydrocortisone valerate	Ointment
	Mometasone furoate	Cream, Lotion, Solution
	Triamcinolone acetonide	Cream, Ointment, Aerosol Spray, Dental Paste
Lower-mid potency (Class 5)	Betamethasone dipropionate	Lotion
	Betamethasone valerate	Cream
	Desonide	Ointment, Gel
	Fluocinolone acetonide	Cream
	Flurandrenolide	Lotion, Cream
	Fluticasone propionate	Lotion
	Hydrocortisone butyrate	Cream, Ointment
	Hydrocortisone probutate	Cream
	Hydrocortisone valerate	Cream
	Prednicarbate	Cream
	Triamcinolone acetonide	Lotion, Ointment
Low potency (Class 6)	Alclometasone dipropionate	Cream, Ointment
	Betamethasone valerate	Lotion
	Desonide	Cream, Lotion, Foam
	Fluocinolone acetonide	Cream, Shampoo, Oil
	Triamcinolone acetonide	Cream
Least potent (Class 7)	Hydrocortisone (base, ≥2%)	Cream, Ointment, Lotion, Solution
	Hydrocortisone (base, <2%)	Ointment, Cream, Gel, Lotion, Spray, Solution
	Hydrocortisone acetate	Cream, Lotion

**Table 2 pharmaceutics-16-00449-t002:** The recently formulated liposomes containing anti-psoriatic agents.

Bioactive Substances	Excipients	Preparation Technique	Particle Size (nm) Polydispersity Index Zeta Potential (mV)	Encapsulation Efficiency (%)	Results of the In Vivo/Ex Vivo Experiments	Ref.
Betamethasone, All-trans-retinoic acid	Lecithin, Tween-80, Ethanol, PBS, Carbomer 940, Glycerin, EDTA, Ethyl paraben, Distilled water, Triethanolamine	Thin-film hydration	65.4 ± 3.1 0.218 ± 0.010 N/A	>98.74 ± 0.78	Reduced epidermal thickness and reduced level of cytokines; a synergistic effect	[84]
Bexarotene	Methyl cellulose, PEG 400, DOTAP, Cholesterol	Thin-film hydration	67.8 ± 7.15 0.26 ± 0.02 46.4 ± 2.8	>90	Reduced psoriatic symptoms supported by positive histopathological, PASI and ELISA reports; 1.8-fold improvement in drug permeation	[85]
Calcipotriol	Cholate, PEG2000-DSPE, Chloroform, Methanol, DSPC, Tris	Thin-film hydration	93.1 ± 0.4 to 95.4 ± 1.5 0.044 ± 0.005 to 0.054 ± 0.010 −4.1 ± 0.1 to −14.4 ± 0.5	N/A	Effective delivery of calcipotriol into excised pig skin; a significant increase in calcipotriol deposition in the stratum corneum	[86]
Cyclosporine	DOTAP, Cholesterol, Chloroform, Carbopol 934, Triethanolamine	Thin-film hydration and ethanol injection method	111 ± 1.62 0.27 ± 0.08 41.12 ± 3.56	93 ± 2.12	Reduced levels of inflammatory markers	[87]
Ginsenoside Rg3	Hyaluronic acid, Lecithin, PBS, Ethanol, Chloroform, Polyvinyl alcohol	Thin-film hydration	110 ± 2.55 0.156 ± 0.03 N/A	80.11 ± 3	Extended duration of drug retention in the skin; improved regulatory effect on cytokines and chemokines, and alleviated symptoms in mice.	[88]
Omiganan	HSPC, DPPC, Soya-lecithin, Cholesterol, Chloroform, PBS, Carbopol 934P, Propylene glycol, Sodium hydroxide	Reverse-phase evaporation	120.5 ± 4.1 0.177 ± 0.012 −17.2	71.89 ± 1.05	Improved cellular uptake in vitro and ex vivo; controlled release and better permeation profile; significant reduction in IL-4, IL-6 and TNF-α levels; improvement in psoriatic lesions.	[89]
Celastrol	Lecithin, Tween 80, Mannose-PEG2000-DSPE, Chloroform, Saline,	Thin-film dispersion	85.5 ± 0.7 0.248 ± 0.01 −1.8 ± 0.2 to −6.0 ± 0.3	90.8 ± 0.4	Inhibited inflammatory cytokines and reduced cytokine storm	[91]
Glabridin	N/A	Fusion method	~100 0.2 N/A	N/A	Reduced PASI scores in mice; reduced epidermal dysplasia, mast cell infiltration, and degranulation; downregulated expression of proinflammatory factors.	[93]
Ibrutinib, Curcumin	Cholesterol, Carbopol 940, Egg lecithin, Phospholipon 90 G, Phospholipon 90 H	Modified solvent injection	128.53 ± 1.18 0.174 ± 0.08 36.16 ± 0.35	83.15 ± 1.91; 86.69 ± 3.80	Anti-inflammatory effect and synergism in psoriasis; reduced pro-inflammatory cytokine levels	[94]

N/A, not available; PASI, Psoriasis Area and Severity Index; PEG, polyethylene glycol; DSPE, distearoyl phosphoethanolamine; DOTAP, 1,2-dioleoyl-3-trimethylammoniumpropane; HSPC, hydrogenated soybean phosphatidylcholine; DPPC, dipalmitoyl phosphatidylcholine; PBS, phosphate buffered saline; EDTA, ethylenediamine tetraacetic acid.

**Table 3 pharmaceutics-16-00449-t003:** The recently formulated solid lipid nanocarriers containing anti-psoriatic agents.

Bioactive Substances	Excipients	Preparation Technique	Particle Size (nm) Polydispersity Index Zeta Potential (mV)	Encapsulation Efficiency (%)	Results of the In Vivo/Ex Vivo Experiments	Ref.
Methotrexate	Glyceryl monostearate, Egg lecithin, Tween-80	Micro-emulsification technique	~253 N/A N/A	85.12	Dose-dependent suppression of keratinocyte expansion; slow, sustained and targeted delivery without being accumulated	[96]
Cyclosporine	Naringenin, Linolenic Acid, Tween 20, Sodium taurocholate, Butanol, Poloxamer 407, Transcutol P, Lecithin, Carbopol, Hyaluronic acid	Micro-emulsification technique	470.0 ± 4.6 0.195 ± 0.047 N/A	92	More advantageous release profile in case of the Poloxamer 407 gel	[98]
Leflunomide	Tween 80, Compritol 888 ATO, Phospholipon G90, Water, Carbopol 934	Micro-emulsification technique	~273.1 0.3 −0.15	65.25 ± 0.95	Significant anti-inflammatory effect; improved photostability; alleviated skin irritation and delayed release	[100]
Apremilast	Precirol ATO-5, Kolliphor CS 12, Carbopol 974P, Triethanolamine, Isopropyl alcohol	Hot-emulsification process	167.70 ± 1.5 0.238 21.37 ± 1.40	63.84 ± 0.93	Improved permeation and skin retention; accumulation in hair follicles and deeper layers; 2-fold higher drug retention in the epidermis and 5-fold higher in the dermis	[101]
Noscapine	Glyceryl distearate (Precirol ATO-5), Tween 80, Poloxamer 188	High-shear homogenization method	245.66 ± 17 0.226 ± 0.09 −35.74 ± 2.59	89.77	Accumulation in the dermis; reduced levels of inflammatory and pro-inflammatory cytokines; reduced IL-17-producing T cells at the site; similar therapeutic effects as clobetasol cream	[102]

N/A, not available.

**Table 4 pharmaceutics-16-00449-t004:** The recently formulated nanostructured lipid carriers containing anti-psoriatic agents.

Bioactive Substances	Excipients	Preparation Technique	Particle Size (nm) Polydispersity Index Zeta Potential (mV)	Encapsulation Efficiency (%)	Results of the In Vivo/Ex Vivo Experiments	Ref.
Luteolin	Ethanol, Glyceryl monostearate, Medium-chain triglycerides, Poloxamer 188, Distilled water, Carbomer 940, Glycerol, Triethanolamine	Melt-emulsification ultrasonic method	199.9 ± 2.6 0.24 ± 0.01 −32.44 ± 0.96	99.81	High penetration and skin retention; accumulation in the hair follicles; decreased psoriasis-like symptoms; reduced PASI scores	[104]
Tacrolimus, Thymoquinone	Capryol 90, Glyceryl monostearate, Dichloromethane, Tween 80, Span 20, Double-distilled water, Mannitol, Carbopol Ultrez 10, Triethanolamine	Emulsifying-solvent evaporation method	144.95 ±2.80 0.160 ± 0.021 −29.47 ± 1.9	>70%	Higher permeation rates in the deeper layers of the skin	[105]
Riluzole	Lutrol F68, Beeswax, Lavender oil, Peppermint oil, Tween 80	Hot-pressure homogenization	192.6 ± 0.8 0.161 ± 0.019 −25.3 ± 0.17	87.16 ± 2.1	Anti-proliferative effects in keratinocytes; a long-lasting slow release; suitable stability and a good ocular tolerance	[106]
Cannabidiol	Cetyl palmitate, Tego Care 450/Poloxamer 188, Transcutol-P, Medium chain triglycerides, Oleic acid, Unigerm G2, Water	Hot high-pressure homogenization	163.2 ± 1.2 to 175.5 ± 1.9 0.009 ± 0.011 to 0.087 ± 0.023 −29.0 ± 0.5 to −57.0 ± 1.7	99.99	Reduced cytotoxicity; enhanced anti-inflammatory activities in macrophages	[107]
Tazarotene, Calcipotriol	Precirol ATO-5, Miglyol 812 N, Chloroform, Pluronic F-188, Distilled water, Carbopol 934	Melt-emulsification technique	149.340 ± 0.340 0.196 ± 0.43 13.14 ± 0.20	91.24 ± 7.30	Less toxicity to HaCaT cells; higher cellular uptake; faster reductions in inflammation in case of combination therapy	[108]
Fluocinolone acetonide, Acitretin	Stearic acid, Oleic acid, Span 80, Tween 80, Carbopol 934, Distilled water, Triethanolamine	Micro-emulsification technique	288.2 ± 2.3 0.345 ± 0.005 −34.2 ± 1.0	75 ± 1.3 for fluocinolone acetonide, 81.6 ± 1.1 for acitretin	Lower permeation rate of both drugs from the coloaded gel; greater skin deposition than the conventional gel	[112]

PASI, Psoriasis Area and Severity Index.

**Table 5 pharmaceutics-16-00449-t005:** The recently formulated nanoemulsions containing anti-psoriatic agents.

Bioactive Substances	Excipients	Preparation Technique	Particle Size (nm) Polydispersity Index Zeta Potential (mV)	Encapsulation Efficiency (%)	Results of the In Vivo/Ex Vivo Experiments	Ref.
Tacrolimus, Azelaic acid	Soy lecithin, E vitamin oil, Chloroform, Cholesterol, Ethanol, PEG 400, Distilled water, Carbopol 940	High-speed homogenization technique	262.6 ± 9.2 0.251 ± 0.03 N/A	N/A	Higher drug retainability	[116]
Fluticasone propionate	Capmul MCM C8, Babchi oil, Cremophor RH 40, Labrafil 1944 CS, Carbopol 980 NF, Aloe vera	Spontaneous emulsification	29.1 ± 21.74 to 56.35 ± 2.83 0.089 ± 0.066 to 0.257 ± 0.015 N/A	N/A	Decreased epidermis thickness; 4-fold increase in skin permeation; higher anti-psoriatic potential compared to the marketed formulation	[118]
Tofacitinib citrate	Eugenoloil, Dimethyl sulfoxide, Tween-20, Transcutol-P, SEPINEO P 600	Spontaneous emulsification	176 ± 11 0.25 ± 0.04 N/A	N/A	Significant reduction in dermatitis score and pro-inflammatory cytokine levels; enhanced in vivo anti-inflammatory activity	[117]
Thymoquinone, Fulvic acid	Kalonji oil, Tween 80, Transcutol-P, Carbopol-971	Aqueous titration method	72.34 ± 2.43 0.126 ± 0.014 −2.83 ± 0.14	N/A	Improved efficacy based on PASI scoring, histopathology and ELISA; no skin irritation	[120]
Curcumin	Labrafac PG, Transcutol HP, Tween 20, Solutol HS15, Carbopol 934	Low-energy emulsification method	10.57 to 68.87 0.094 to 0.550 −3.90 to −18.7	N/A	Faster improvement of the psoriatic signs investigated in mice and compared with the gel without nanoemulsion	[122]
Curcumin, Resveratrol, Thymoquinone	Oleic acid, Tween 20, PEG 200, Carbopol 940, Distilled water	Aqueous titration method	76.20 ± 1.67 to 324.7 ± 1.20 0.12 ± 0.05 to 0.32 ± 0.04N/A	N/A	Higher drug deposition in the skin; significant reduction in inflammation and scaly lesions; disappearance of hyperkeratosis and parakeratosis of psoriatic skin	[123]
Babchi oil	Rhamnolipid, Propylene glycol, Carbomer 940, Distilled water	Low-energy emulsification method	120.34 to 302.45 0.29 ± 0.073 to 0.67 ± 0.060 −12.76 to −29.08	58.21 to 93.12	Significant improvement in drug permeability	[119]
Methotrexate	Olive oil, PEG 400, Tween 80, Distilled water, Sodium alginate, Triethanolamine	High-shear homogenization technique	202.6 ± 11.59 0.233 ± 0.01 −14.2 ± 4.42	76.57 ± 2.48	Increased skin penetration; decrease in PASI score	[124]
Tacrolimus	Fish oil, Linseed oil, Distilled water, Tween 80, Transcutol-P, Carbopol-934	Spontaneous emulsification and high-pressure homogenization	116.3 ± 20.8; 126.3 ± 27.1 0.178 ± 0.011; 0.194 ± 0.008 −3.99 ± 0.17; −3.13 ± 0.11	N/A	High anti-psoriatic activity in mice; reduced cytokine levels in the skin	[125]

N/A, not available; PASI, Psoriasis Area and Severity Index; PEG, polyethylene glycol.

**Table 6 pharmaceutics-16-00449-t006:** The recently formulated ethosomes containing anti-psoriatic agents.

Bioactive Substances	Excipients	Preparation Technique	Particle Size (nm) Polydispersity Index Zeta Potential (mV)	Encapsulation Efficiency (%)	Results of the In Vivo/Ex Vivo Experiments	Ref.
Apremilast	Cholesterol, Soy lecithin, Propylene glycol, Ethanol, Water, Carbopol 340, Triethanolamine, Benzalkonium chloride	One-step injection technique	111 ± 22 <0.3 −56.3	81.29 ± 5.45	Higher bioavailability than the orally administered suspension	[127]
Tacrolimus, Hyaluronic acid	Soya lecithin, Propylene glycol, Ethanol, Distilled water, Carbopol 934, Triethanolamine	Ethanol injection method	315.7 ± 2.2 0.472 ± 0.07 −18.5 ± 1.2	88.3 ± 2.52	Controlled drug release; sustained drug availability at the application site; decreased edema in rats	[129]
Resveratrol	Soya phosphatidylcholine, Stearyl amine, Propylene glycol, Carbopol 934P, Triethanolamine, Ethanol, Double-distilled water	Cold method	196.8 ± 4.19 0.221 ± 0.013 21.31 ± 1.53	41.27 ± 1.43 to 71.83 ± 1.22	Adequate penetration into the deeper skin layers	[131]
Curcumin	DSPE-PEG2000, Hydrogenated soybean phospholipids, Cholesterol, Propylene glycol, Hyaluronic acid	Aqueous titration method	~195 <0.3 ~32	~90	Reduced inflammation symptoms; greater downregulation in the proinflammatory cytokine levels	[130]
Mangiferin	Lipoid S75, Tween 80, Glycerol, Ethanol, Water	Hot-homogenization method followed by ultrasound	~145 0.4 ~−42 ± 1	~69 ± 1	Significantly reduced TPA-induced oedema; additional wound-healing activity	[132]
Thymoquinone	Phospholipon 90 G, Ethanol, Distilled water, Carbopol 934	Cold method	253.7 to 531.3 N/A N/A	45.00 ± 0.75 to 79.52 ± 0.24	Improved retention of thymoquinone in the dermal layers; enhanced anti-psoriatic activity	[133]

N/A, not available; DSPE-PEG2000, 1,2-distearoyl-sn-glycero-3-phosphoethanolamine-N-[amino(polyethylene glycol)-2000] (ammonium salt); TPA, tetradecanoyl phorbol acetate.

**Table 7 pharmaceutics-16-00449-t007:** The recently formulated niosomes containing anti-psoriatic agents.

Bioactive Substances	Excipients	Preparation Technique	Particle Size (nm) Polydispersity Index Zeta Potential (mV)	Encapsulation Efficiency (%)	Results of the In Vivo/Ex Vivo Experiments	Ref.
Celastrol	Cholesterol, Span 20, Span 60, Chloroform, Methanol, Carbopol 974	Thin-film hydration, sonication techniques	147.4 ± 5.6 0.258 ± 0.02 −48.9 ± 1.1	N/A	Reduced erythema and scaling of the skin in mouse models; decreased cytokine levels	[140]
Cyclosporine	Cholesterol, Span 60, Carbopol 940, Methanol, Chloroform	Thin-film hydration	100.0 ± 0.8 to 284.5 ± 1.6 0.12 ± 0.02 to 0.31 ± 0.04 −32.2 ± 0.9 to 57.0 ± 0.9	88.23 to 96.5	Higher permeation in comparison to the cyclosporine suspension; 59-fold increase in cyclosporine deposition in the stratum corneum and in the epidermis/dermis layers; a significant decrease in PASI scores	[141]
Cyclosporine, Pentoxifylline	Cholesterol, Tween 80, Span 80, Chloroform, Methanol,	Thin-film hydration	~179 ~0.285 ~−37.5	~75.03	Improved permeation of both cyclosporine and pentoxifylline; marked improvement in the skin condition of mice according to the histopathology and PASI scores	[143]
Pentoxifylline	Cholesterol, Tween 80, Soya lecithin, Chloroform, Methanol	Thin-film hydration	69.52 to 1340.7 0.019 to 0.338 22.7 to 38.6	21.98 to 77.23	Reduced inflammation of epidermis and stratum corneum; reduction in the PASI scores	[134]
Cyclosporine	Cholesterol, Span 60, Methanol, Chloroform,	Thin-film hydration	61.96 ± 0.91 to 193.7 ± 0.23 0.101 ± 0.024 to 0.587 ± 0.098 −20.7 to 31.6	80.53 ± 1.18 to 98.19 ± 0.94	Deeper and greater accumulation; improvement in the skin condition of mice according to the histopathology and PASI scores compared to the drug dispersion.	[144]

N/A, not available; PASI, Psoriasis Area and Severity Index.

**Table 8 pharmaceutics-16-00449-t008:** The recently formulated nanosponges containing anti-psoriatic agents.

Bioactive Substances	Excipients	Preparation Technique	Particle Size (nm) Polydispersity Index Zeta Potential (mV)	Encapsulation Efficiency (%)	Results of the In Vivo/Ex Vivo Experiments	Ref.
Curcumin, Caffeine	β-cyclodextrin, Dimethyl carbonate, Ethanol, Water, Carbopol-934, Triethanolamine, Guar gum, Propylene glycol	Hot-melt method	170 to 200 0.291 ± 0.073 to 0.395 ± 0.026 14.6 ± 1.1 to −28.35	50.26 to 61.14	Reduction in epidermal thickness; sustained release	[146]
Clobetasol propionate	β-cyclodextrin, Diphenyl carbonate, Methanol, Double-distilled water, Acetone, Carbopol 934	Hot-melt method	194.27 ± 49.24 0.026 −21.83 ± 0.95	56.33 ± 0.94	Reduction in epidermal thickness; enhanced anti-psoriatic potential	[147]
Dithranol	β-cyclodextrin, Diphenyl carbonate, Carbopol 934, Distilled water, Acetone, Triethanolamine	Melt method	274.6 ± 43.54 0.545 −28.3 ± 6.34	N/A	Reduction in epidermal thickness investigated in a mouse tail model	[149]
	β-cyclodextrin, Diphenyl carbonate, Dimethylformamide, Distilled water, Acetone	Solvent evaporation technique	373.8 ± 21.2 to 978.35 ± 88.8 0.537 ± 0.193 to 0.833 ± 0.236 −15.0 ± 2.4 to −23.9 ± 1.2	58.77 ± 0.53 to 83.59 ± 1.45	Improved solubility and photostability; significant antioxidant efficacy	[148]

N/A, not available.

**Table 9 pharmaceutics-16-00449-t009:** The recently formulated dendrimers containing anti-psoriatic agents.

Bioactive Substances	Excipients	Preparation Technique	Particle Size (nm) Polydispersity Index Zeta Potential (mV)	Encapsulation Efficiency (%)	Results of the In Vivo/Ex Vivo Experiments	Ref.
Dithranol	PAMAM dendrimer, Ethyl cellulose, Dichloromethane, PVA, Sodium meta-bisulfate, Distilled water	Divergent synthesis and quasi-emulsion solvent diffusion method	28 ± 1.12 to 130 ± 1.01 N/A ~−22	49.21 ± 4.949 to 71.33 ± 5.451	Prolonged skin penetration rate compared to the marketed form; non-irritating when administered to the skin of the rats	[152]
	PPI dendrimer, Ethylenediamine, Acrylonitrile,	Divergent synthesis	8.00 ± 0.04 0.88 ± 0.023 12.0 ± 0.42	57.1 ± 1.32% (at 1.2 pH)	Improved skin penetration rate and enhanced drug-transport in the rat skin	[153]
Isotretinoin	4-dimethylaminopyridine, 3-chloro- 2-chloromethyl-1-propene, Tri-ethylene glycol monomethyl ether, Dicyclohexylcarbodiimide, Toluene, Ethyl acetate, Water	Convergent synthesis and self-assembly	~25 N/A N/A	N/A	Better skin penetration; targeted delivery; minimal skin irritation	[154]

N/A, not available; PAMAM, polyamide amine; PPI, polypropylene imine; PVA, polyvinyl alcohol.

**Table 10 pharmaceutics-16-00449-t010:** The recently formulated nanocrystals containing anti-psoriatic agents.

Bioactive Substances	Excipients	Preparation Technique	Particle Size (nm) Polydispersity Index Zeta Potential (mV)	Encapsulation Efficiency (%)	Results of the In Vivo/Ex Vivo Experiments	Ref.
18β-glycyrrhetinic acid	Sodium dodecyl sulfate, Carbomer 940, Ethanol, Water	High-pressure homogenization, freeze-drying	288.6 ± 7.3 0.13 ± 0.10 −45.9 ± 1.3	N/A	Higher cumulative permeation; a significantly greater inhibitory effect on TPA-induced inflammation	[156]
Curcumin	Polyvinylpyrrolidone, Xanthan gum, Propylene glycol	Anti-solvent precipitation technique	60, 120, 480 <0.3 N/A	N/A	Higher accumulation, permeation, and skin retention after 24 h	[155]
Indirubin	Hyaluronic acid, Poloxamer 188, Pluronic F127, TPGS, PEG 4000, Tween 80	Wet-media milling technique	229.6 ± 3.0 0.21 ± 0.02 1.4 ± 0.002	N/A	Increased accumulation in inflammatory lesions; higher downregulation of inflammatory cytokines	[157]
Diosmin	Dimethyl sulfoxide, HPMC E15, Methyl cellulose, Poloxamer 407, Sodium alginate, Water	Anti-solvent precipitation technique	276.9 ± 16.49 0.43 ± 0.02 N/A	N/A	Significantly reduced epidermal thickness; dose-dependent efficacy	[158]
Rutin	Pluronic F-17, Tween 80, HP-β-CD, PEG 6000, PEG 200, Ethanol, Water	Anti-solvent nanoprecipitation-ultrasonic treatment technique	270.5 ± 16.7 to 505.8 ± 20.5 0.32 ± 0.02 to 0.56 ± 0.7 −12.4 ± 1.0 to−28.8 ± 1.0	65.7 ± 0.7 to 75.5 ± 0.9	Inhibited edema	[159]
*Gelidium amansii*	Cellulose (Avicel^®^ pH-101), Distilled water, Sulfuric acid, NaOH	Acid-hydrolyzation, purification	307.38 ± 114.25 nm length and 18.95 ± 6.87 width N/A N/A	N/A	Significant inhibition of UVB-induced COX-2 expression is mouse skin	[160]

N/A, not available; TPA, tetradecanoyl phorbol acetate; COX-2, cyclooxygenase-2; TPGS, α-D-tocopheryl polyethylene glycol 1000 succinate; PEG, polyethylene glycol; HPMC, hydroxypropyl methyl cellulose, HP-β-CD, hydroxypropyl-β-cyclodextrin.

## Data Availability

Not applicable.

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
