# Peer review of "Recent Approaches for the Topical Treatment of Psoriasis Using Nanoparticles"

_pharmaceutics, 2024, doi:10.3390/pharmaceutics16040449_

Round 1

Reviewer 1 Report

Comments and Suggestions for Authors

The manuscript pharmaceutics-2894063 “Recent approaches for the topical treatment of psoriasis using nanoparticles” by Krisztina Bodnár et al. reviews a novel approach to treat psoriasis using nanoparticles to increase therapeutic efficacy and reduce the toxicity profile of drugs. 

The manuscript is well written and structured and contains up-to-date information. The used references are reasonable.

Questions and comments:

  1. Line 108 - 128 - The pathogenesis of psoriasis should be presented as a figure.

  2. Also, information about the structure of the skin should be added briefly. 

  3. Section 3 - Disadvantages of traditional dosage forms for topical application should be highlighted.

  4. Lines 247 - 249 - It should be noted that polymeric drug delivery systems can be larger than 100 nm (submicron particles); polymer scientists traditionally refer to them as drug nanocarriers.

  5. Section 6.1. - Problems of liposome stability during storage should be noted.

  6. Polyelectrolyte particles for topical treatment of psoriasis may be a promising nanotechnology-based dosage form.  For example, polyelectrolyte complexes based on chitosan and fucoidan. Some references may be helpful to you, for instance, doi.org/10.3390/ijms24032615

doi.org/10.1016/j.carbpol.2021.118238

Author Response

Dear Reviewer 1,

I hereby submit our modified manuscript “Recent approaches for the topical treatment of psoriasis using nanoparticles”, by Krisztina Bodnár et al. Thank you for your constructive comments, according to them, the following changes were made during revision (corrections related to the Reviewer 1 are marked with yellow in the manuscript):

  1. A figure introducing the pathomechanism of psoriasis was added to the manuscript (page 3., Figure 1.)
  2. Information about the structure of the skin was also added to the beginning of the section 2. (lines 84-97)
  3. In section 3 the disadvantages of the traditional dosage forms for topical application were highlighted, as it was advised. (lines 231-240)
  4. As it was suggested, it was that polymeric drug delivery systems could be larger than 100 nm (submicron particles) (lines 277-280): Polymer scientists often refer to polymer drug delivery systems as "drug nanocarriers" even though their size can exceed 100 nm. This term reflects their primary function of carrying drugs within a nanoscale structure, facilitating targeted delivery, and improving drug efficacy.”
  5. The main problems of liposome stability during storage should were added (lines 391-394): „The primary issue encountered when using liposomes lies in their limited physical and chemical stability, attributed to the delicate nature of phospholipid membranes and their susceptibility to peroxidation. Physical degradation may also occur because of the alterations in the structure of the liposomes.”
  6. The manuscript was supplemented with a new chapter (6.10) in which we wrote about polyelectrolyte nanoparticles. The suggested references were also added to the manuscript.

We hope that our improved manuscript is suitable for publication.

Waiting for your kind response.

Sincerely,

Liza Józsa

corresponding author

Reviewer 2 Report

Comments and Suggestions for Authors

Manuscript "Recent approaches for the topical treatment of psoriasis using nanoparticles" shows important advances on the treatment of psoriasis with nanoparticles. However, the authors should expand the manuscript by adding the sections on Transfersomes, Polymersomes, Emulsomes, Solid-phase carriers Gold nanoparticles, Solid polyemeric nanoparticles, Nanogels.
Likewise, the authors could mention which nanoparticles are better for different types of psoriasis (Plaque psoriasis, Guttate psoriasis, Pustular psoriasis, Inverse psoriasis).

Comments on the Quality of English Language

Authors should review the entire manuscript, as there are verbs in different tenses.

Author Response

Dear Reviewer 2,

I hereby submit our modified manuscript “Recent approaches for the topical treatment of psoriasis using nanoparticles”, by Krisztina Bodnár et al. Thank you for your constructive comments, according to them, the following changes were made during revision (corrections related to the Reviewer 2 are marked with green in the manuscript):

The manuscript was expanded by adding sections on Polymersomes (6.11), Transfersomes (6.12), Emulsomes (6.13), Gold nanoparticles (6.14), Solid polymeric nanoparticles (6.15), and Nanogels (6.16), as advised. The latest research results related to psoriasis were summarized in the mentioned chapters.

Another chapter (Chapter 8: Nanoparticle Selection according to the Type of Psoriasis) was added to the manuscript, detailing the optimal nanoparticles for each type of psoriasis.

The entire manuscript was revised, and the verb tense usage was corrected.

We hope that our improved manuscript is suitable for publication.

Waiting for your kind response.

Sincerely,

Liza Józsa

corresponding author

Reviewer 3 Report

Comments and Suggestions for Authors

I have reviewed the manuscript "Recent approaches for the topical treatment of psoriasis using nanoparticles". The review article presents relevant information about the results of the in vitro and in vivo examinations carried out in the last few years regarding the effectiveness of nanoparticles in psoriasis treatment. This manuscript is interesting, well-written, and adequately described. However, I recommend some minor revisions before considering it for publication:

-        The description of the main findings of the studies presented in the article is repeated twice, both in the text and in the table, resulting in an unnecessarily lengthy paper. The duplication of studies should be avoided, and only one method of description should be chosen. It could be a combination, with some studies detailed in the text and others presented in a table or specifying one approach for describing one type of nanostructure and using a table for another.

-        Although briefly mentioned, the authors should provide a more detailed description of the limitations of using nanotechnology, such as potential toxicity, allergies, degradation in the body, costs, and regulatory considerations.

Author Response

Dear Reviewer 3,

I hereby submit our modified manuscript “Recent approaches for the topical treatment of psoriasis using nanoparticles”, by Krisztina Bodnár et al. Thank you for your positive evaluation and constructive comments, according to them, the following changes were made during revision (corrections related to the Reviewer 3 are marked with blue in the manuscript):

  1. The descriptions of the main findings were shortened. The information that is also included in the table was removed from the text. However, we were unable to shorten the length of the article as the other reviewers requested the addition of new chapters.

  2. A more detailed description of the limitations of using nanotechnology was added to chapter 5 (lines 345 – 366)

We hope that our improved manuscript is suitable for publication.

Waiting for your kind response.

Sincerely,

Liza Józsa

corresponding author

Round 2

Reviewer 1 Report

Comments and Suggestions for Authors

The paper may be accepted.

Reviewer 2 Report

Comments and Suggestions for Authors

The authors of manuscript "Recent approaches for the topical treatment of psoriasis using nanoparticles" have made the relevant corrections suggested by the reviewers. The manuscript in its current state can be published on the journal's platform.